

# Improved basal drag of the West Antarctic Ice Sheet from L-curve analysis of inverse models utilizing subglacial hydrology simulations

Lea-Sophie Höyns[1,2], Thomas Kleiner[1], Andreas Rademacher[2], Martin Rückamp[3], Michael Wolovick[1], and Angelika Humbert[1,4]

[1]Alfred-Wegener Institute, Helmholtz Centre for Polar and Marine Research, Section Glaciology, Bremerhaven, Germany
[2]University of Bremen, Center for Industrial Mathematics, Bremen, Germany
[3]Bavarian Academy of Sciences and Humanities, Geodesy and Glaciology, Munich, Germany
[4]University of Bremen, Department of Geosciences, Bremen, Germany

**Correspondence:** Lea-Sophie Höyns (lea-sophie.hoeyns@awi.de)

**Abstract.**

The West Antarctic Ice Sheet (WAIS) is the focus of current research due to its susceptibility to collapse, which could potentially contribute to rising sea levels. To accurately predict future glacier evolution, precise ice sheet models are essential with regard to suitable approximations of physical behavior to the real system and appropriate input values, as well as computing power. The ice discharge of outlet glaciers into the ocean is one key factor here, primarily caused by basal sliding of ice. Since we cannot directly measure basal properties on a large scale, inverse models can be used to infer the basal drag coefficient by minimizing a cost function that depends on a velocity misfit and a regularization term.

We conduct basal drag inversions and perform L-curve analyses to find the optimal trade-off between the cost function terms, ending up with smooth L-curves. Additionally, the domain L-curve is divided into eight subdomains of the study area in order to reveal how well the inverse method performs in different glaciological settings. It reveals that Pine Island Glacier being the best area, and slow-flowing areas such as Roosevelt Island being among the worst in terms of the L-curve behavior for the basal drag inversion. This highlights the importance of performing a subdomain L-curve analysis, whenever an inversion for a larger domain is calculated to discover problematic regions. Comprehensive basal drag inversion experiments allow us to test the dependence of the L-curve and basal drag results on the non-linearity of sliding as well as on the inclusion of subglacial effective pressure in the friction law. The analysis suggests that non-linear friction laws are preferable to linear sliding because of reduced variance of the overall inferred friction coefficient, faster convergence, as well as steeper L-curves leading to a more accurate choice of weight for the regularization term. We show that a Budd-type friction law that incorporates effective pressure from a subglacial hydrology model rather than a simple geometry-based approximation achieves an improved performance in our inverse model. Further comparison reveals that the effective pressure from the hydrology model accounts for a larger part of the spatial basal drag coefficient structure than the parameterized effective pressure. Allowing the inverted drag coefficient to more precisely reflect actual variations in basal properties. Finally, a comparison of the inferred basal drag across WAIS with observed locations of subglacial lakes reveals a good match, giving us additional confidence in the spatial basal drag structure revealed by the inverse method.





Keywords: Basal Drag Inversion; Optimization Problems; Data Assimilation; Ice Sheet Models; Higher Order Model; West Antarctic Ice Sheet; Friction Laws; Subglacial Processes; Subglacial Hydrology (Effective Pressure);

# 1 Introduction

The West Antarctic Ice Sheet (WAIS) experiences massive ice loss and is currently the major contributor of Antarctica to sea-level change (Shepherd et al., 2018; Naughten et al., 2023). The instability of WAIS and the related ongoing melt may lead to

a future global sea level rise of about $3.3\,\text{m}$ (Bamber et al., 2009). The ongoing continuous improvement of ice sheet models (e.g., Blatter et al., 2010; Seroussi et al., 2019) likely improves the uncertainties in future predictions involved. This allows us to gain better insights into the mechanisms behind the behavior of ice sheets, e.g., in the ongoing melting and the ice dynamics. The continuous improvements lead to better initial states of ice sheet models and their related uncertainties in future projections (Seroussi et al., 2019). In this context of a better understanding of ice sheet processes, it is particularly important to examine the

distribution of friction at the ice-bed interface, as this process has a major influence on the ice flow velocity. The reason is that the majority of the high velocities of a glacier are caused by sliding along the bed of the glacier. Since the distribution of friction underneath the ice sheets is difficult to observe directly, we need ice flow models to determine a realistic basal drag. The motion of the glacier due to sliding at the base is also strongly linked to the subglacial hydrology (Cuffey and Paterson, 2010; Benn and Evans, 2010), as, simply put, the occurrence of water lubricates the bed. It is therefore desirable to resemble the subglacial

water pressure on a more realistic and physical basis than it has been done so far in the community (e.g., Arthern et al., 2015; Barnes and Gudmundsson, 2022; Kazmierczak et al., 2022). However, since remote sensing data such as surface velocity are available, the problem of determining the basal drag can be mathematically identified as an inverse problem. Solving inverse problems using an optimal control approach can help to compute unknown parameters. The application of inversions to infer the basal drag coefficient is a common approach in the glaciology community (MacAyeal, 1993; Joughin et al., 2004, 2009;

Morlighem et al., 2010, 2013; Habermann et al., 2012; Sergienko and Hindmarsh, 2013; Sergienko et al., 2014; Zhao et al., 2018; Wolovick et al., 2023).

The disadvantage of inverse methods is the instability, as all existing errors in the system are reflected in the unknown basal drag coefficient to be determined, which leads to an inaccurate result. These errors can, for example, be based on incorrect model physics, such as assumptions for the flow law regarding anisotropy, which can influence the ratio of deformation to

sliding flow, as described in McCormack et al. (2022). Rathmann and Lilien (2022) point out, that neglecting the crystal-orientation fabric in the flow law can influence the inferred basal drag coefficient, which can be remedied by including an isotropic enhancement factor and inverting both the rheology and the basal drag coefficient. In addition, the basal drag coefficient is sensitive to temperature assumptions and thus to the determination of ice rheology (Zhao et al., 2018). Kyrke-Smith et al. (2018) find that the accuracy of the bed elevation data affects the derived basal conditions and suggest to invert for both,

the basal drag coefficient and the basal topography. The importance of friction laws and thus the capture of subglacial hydrology contribute to a more realistic basal drag coefficient determined from inversions (Schroeder et al., 2013; McArthur et al., 2023). Overall, there are many assumptions behind every inversion of basal drag, as, for example, there are not necessarily





enough data available or the aim of the studies differ. However, it would be possible to include all the methods mentioned in the inversion procedure presented. But, in this manuscript, we focus primarily on the choice of the friction law and the associated subglacial hydrology to reduce uncertainty in the resulting basal drag coefficient.

To account for sliding beneath a glacier, a friction law (Weertman, 1957) is applied at the ice-base boundary condition of ice sheet models. The accuracy of the unconstrained parameters in this law plays an important role in modeling glaciers in the most realistic sense. In general, the friction law describes the basal drag in terms of basal velocity, a basal drag coefficient, and a description of subglacial hydrology, which enters the law due to effective (water-)pressure (Budd et al., 1979). These parameters are some of the least constrained inputs in ice flow models. In order to overcome the difficulties with unconstrained parameters the inverse method is used to determine the basal drag coefficient. However, since the effective pressure is also included in the friction law, it is crucial to find a good representation of the linked subglacial hydrology in order to separate basal properties that stem from the subglacial hydrology and 'other' properties. Here, we compare a commonly used parameterized effective pressure (e.g., McArthur et al., 2023; Wolovick et al., 2023) with an effective pressure from a subglacial hydrology model (Sect. 2, Subglacial hydrology) to demonstrate the relevance of an improved water pressure description.

In the inversion process, the basal drag coefficient is controlled by minimizing the misfit between simulated and observed surface velocities. The integration of a regularization term (Tikhonov and Arsenin, 1977) in the cost function of the basal drag inversion ensures that unrealistic structures in the solution, which arise due to the ill-posedness of the problem, are smoothed by penalizing oscillations in the basal drag coefficient. In order to achieve a trade-off between the two cost function terms, it is necessary to determine a weight for the regularization term with the help of an L-curve (Hansen, 1992; Hansen and O'Leary, 1993; Hansen, 2001; Wolovick et al., 2023). For this purpose, we follow Wolovick et al. (2023) and perform an L-curve analysis that identifies the best weight for the regularized cost function term at the maximum curvature of the resulting L-shaped curve. In the literature of the glaciological community, the regularization and the related L-curve analysis are not always applied, when an inversion is performed (e.g., MacAyeal, 1993; Joughin et al., 2004, 2009; Arthern and Gudmundsson, 2010). Further, we are not aware of any literature in which the basal drag inversion for the entire WAIS region (compare Fig. 1) is performed using a regularization term, as well as an explicit L-curve analysis (e.g., Joughin et al., 2004, 2009; Ranganathan et al., 2021). In addition, previous studies usually consider only individual glaciers or regions of WAIS, such as Joughin et al. (2004), which focuses on the Ross Ice Shelf, or Ranganathan et al. (2021) concentrating on the MacAyeal Ice Stream, as well as Morlighem et al. (2010) and Gillet-Chaulet et al. (2016) who deal with Pine Island Glacier, and Joughin et al. (2009) and Sergienko and Hindmarsh (2013) who examine both, the Pine Island Glacier and the Thwaites Glacier. Although Morlighem et al. (2013) and Arthern et al. (2015) model the whole Antarctic ice sheet, the results of those studies are nevertheless not based on a high-resolution mesh. In the literature, a Weertman friction law (Weertman, 1957) is often used (e.g., Morlighem et al., 2010, 2013; Joughin et al., 2004; Ranganathan et al., 2021) instead of a Budd-type friction law (Budd et al., 1979), in which no effective pressure is taken into account. It is also common, when using a Budd-type friction law, to use a simple geometry-based parameterization for the effective pressure (e.g., Arthern et al., 2015; Barnes and Gudmundsson, 2022; Kazmierczak et al., 2022). As this parameterization is not ideal due to its strong simplifications on the subglacial processes (e.g., a perfect



connection to the ocean of marine parts of the ice sheet), it is desirable to leverage results of hydrology models (e.g., Koziol and Arnold, 2017; Beyer et al., 2018; Gilbert et al., 2022; McArthur et al., 2023).

One objective of this paper is to test whether an improved description of the effective pressure results in a more reliable basal drag distribution for a major part of WAIS. Therefore we leverage the effective pressure from a physical-based subglacial hydrology model in the friction law and apply the common basal drag inversion. We use the effective pressure of the confined-unconfined aquifer system model (CUAS-MPI; Beyer et al. (2018); Fischler et al. (2023)) as it was shown to perform well in SHMIP (De Fleurian et al., 2018) and is able to describe different states of the water system. In addition, we conduct a subdomain L-curve analysis in order to explore how the well-posedness of the inverse problem varies with glaciological

settings. Based on the performed simulations, we examine the basal drag distribution and analyze the influence of the different effective pressure maps, as well as the linear and non-linear friction law on the L-curve and on the spatial variability of the basal drag coefficient.

In the following paper, we first describe the methods and data that we use to perform the inversion for the WAIS (Sect. 2). We present our results regarding the spatial distribution of the basal drag and the basal drag coefficient along with the obtained L-

curves and the subdomain L-curve analysis (Sect. 3). Finally, we discuss our findings and compare them with lake candidates, as well as other studies (Sect. 4).

## 2    Method

Our simulations are conducted within the open-source, finite-element based Ice-Sheet and Sea-level System Model (ISSM; Larour et al. (2012)). The basis of our inversion approach is build through the model equations represented by an ice flow

model. Completed with its boundary conditions, we refer to it as our forward model. The unknown basal drag coefficient to be controlled by the inversion is contained in the boundary conditions at the ice-base interface. The latter is represented through both, a (non-)linear Weertman-type (Weertman, 1957), as well as by a Budd-type friction law (Budd et al., 1979) using different effective pressure fields.

In the following subsections, we describe the ice flow model setup for the study region covering the WAIS. Subsequently,

we present the forward model, as well as the inversion process including regularization and the L-curve analysis.

### 2.1   Model setup

In total, we perform six basal drag inversions with an accompanying L-curve experiment (compare Sect. 3). The experiments encompass setups with linear and non-linear sliding for both, Weertman- and Budd-type friction laws. To review the effect of different effective pressure fields, we either set this field to an effective pressure from the subglacial hydrology of CUAS-MPI,

to a simple geometry-based parameterization or we even neglect the effective pressure entirely (Weertman friction law).





## Application to the West Antarctic Ice Sheet

We choose the WAIS domain (Fig. 1) by using the defined ice sheet drainage basins of Rignot (Glovinetto and Zwally, 2000; Rignot et al., 2011a, c, 2013) from the Ice Sheet Mass Balance Inter-comparison Exercise (IMBIE-3, Rignot et al. (2019)). As the ice shelves are not included in those basins, we include them with the MEaSUREs Antarctic Boundaries dataset (Mouginot

et al., 2017). We exclude the so-called *J-Jpp* basin describing the Filchner-Ronne catchment to keep the computational effort on a manageable level. However, results of the basal drag of the J-Jpp basin have already been published by Wolovick et al. (2023), which we do not aim to reproduce. We simulate the basins for Marie Byrd Land and Ellsworth Land without the Weddell Sea Sector. For the sake of simplicity, we will nevertheless refer to it as WAIS in the following. Figure 1 illustrates

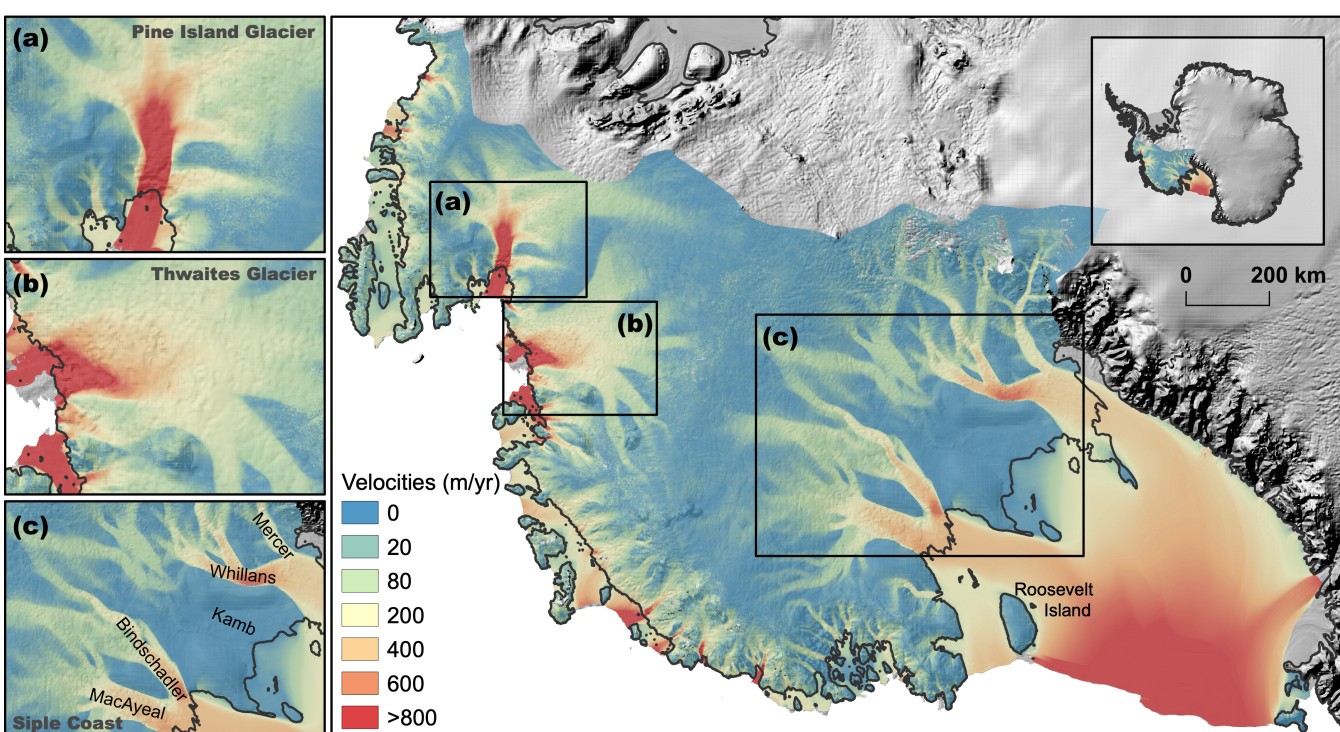

**Figure 1.** Map of the study domain covering a large part of WAIS. The plots show the observed surface velocities in $\mathrm{m\,yr^{-1}}$ from the MEaSUREsv2 dataset (Rignot et al., 2011b, 2017). The background imagery displays the ice surface elevation of Antarctica from the BedMachine Antarctica v2 dataset (Morlighem et al., 2020; Morlighem, 2020). The black line represents the grounding line. The insets (a), (b) and (c) show zooms to Pine Island Glacier, Thwaites Glacier and the Siple Coast with the Mercer, Whillans, Kamb, Bindschadler and MayAyeal Ice Streams, respectively

that we are dealing with both slow- and fast-flowing areas in this domain, including Thwaites and Pine Island Glaciers as well

as the Siple Coast ice streams. In Fig. 2a the bed elevation is displayed, which clearly shows that most of the WAIS area lies below sea level, with the interior deeper than the margins, making it vulnerable to the marine ice sheet instability (e.g., Hughes,



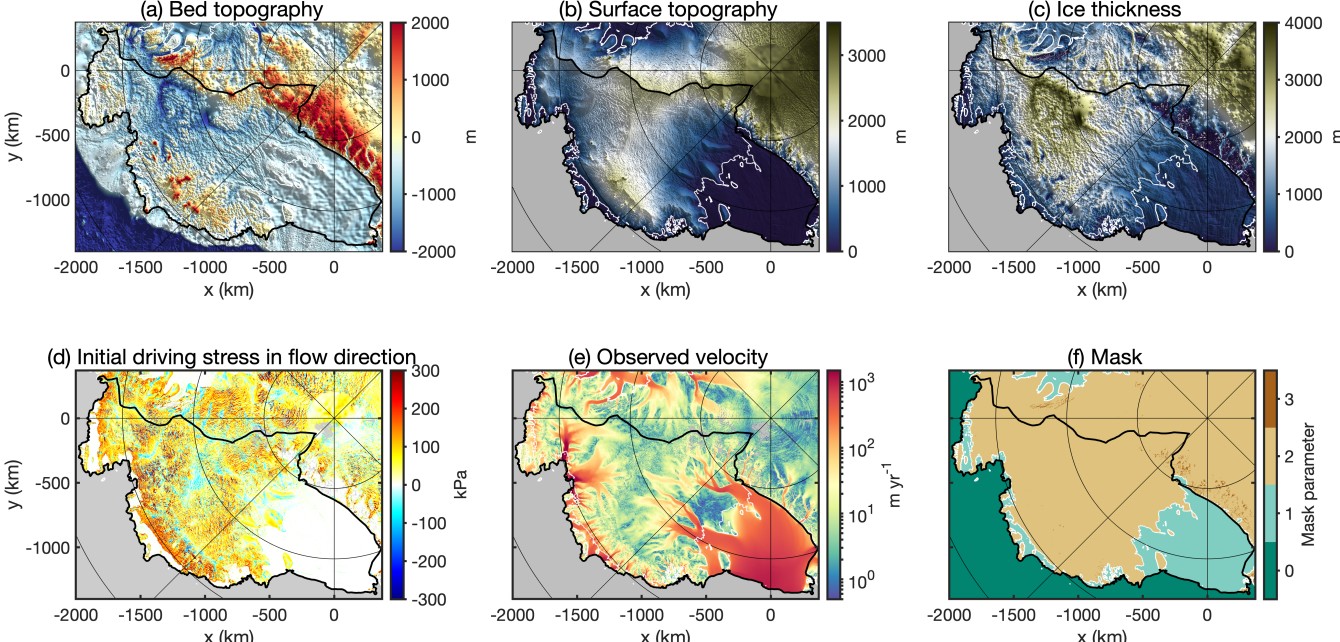

**Figure 2.** Model setup of WAIS domain. (a) Bed topography in $\mathrm{m}$. (b) Surface elevation in $\mathrm{m}$. (c) Ice thickness in $\mathrm{m}$. (d) The initial driving stress in flow direction in kPa. (e) Observed surface velocity in $\mathrm{m\,yr^{-1}}$ in log-scale from the MEaSUREsv2 dataset (Mouginot et al., 2012; Rignot et al., 2011b, 2017). (f) WAIS mask with the mask parameter 0 describes the ocean, 1 the floating ice, 2 the grounded ice and 3 describes exposed rock. The subplots (a–d and f) are based on the BedMachine Antarctica v2 dataset (Morlighem et al., 2020; Morlighem, 2020). The white line in the subplots describes the grounding line and the black line delineates the study area. The plots (a–c) are underlain with a hillshade. Gray areas in (d–e) indicate regions with no available data.

1973; Weertman, 1974; Thomas and Bentley, 1978; Schoof, 2007). The black line in Fig. 1 delineates the grounding line. The geometry is based on the BedMachine Antarctica v2 dataset (Morlighem et al., 2020; Morlighem, 2020). This dataset includes bed topography (Fig. 2a), surface elevation (Fig. 2b), as well as ice thickness (Fig. 2c) and the mask for the WAIS (Fig. 2f). As

described in Sect. 2.3 the most relevant data for the inversion procedure to fit the modeled horizontal ice velocities $v_x, v_y$ are the observed ice surface velocities $\boldsymbol{v}_\mathrm{s}^\mathrm{obs}$. Here we use ice surface velocities from the MEaSUREs v2 dataset (Mouginot et al., 2012; Rignot et al., 2011b, 2017) as target in the inversion as it has a complete spatial coverage in the modeling domain (Fig. 2e).

**Mesh construction**

We construct a two-dimensional unstructured, triangular mesh generated by using the Bidimensional Anisotropic Mesh Generator (BAMG, Hecht (2006)). The horizontal mesh is refined in dynamic active regions such as the shear margins (Fig. 3a), the calving front, as well as in fast-flowing areas and outlet glaciers and at the grounding line. The mesh is extruded into 10 vertical layers, so that we get a three-dimensional prism structure of the mesh.



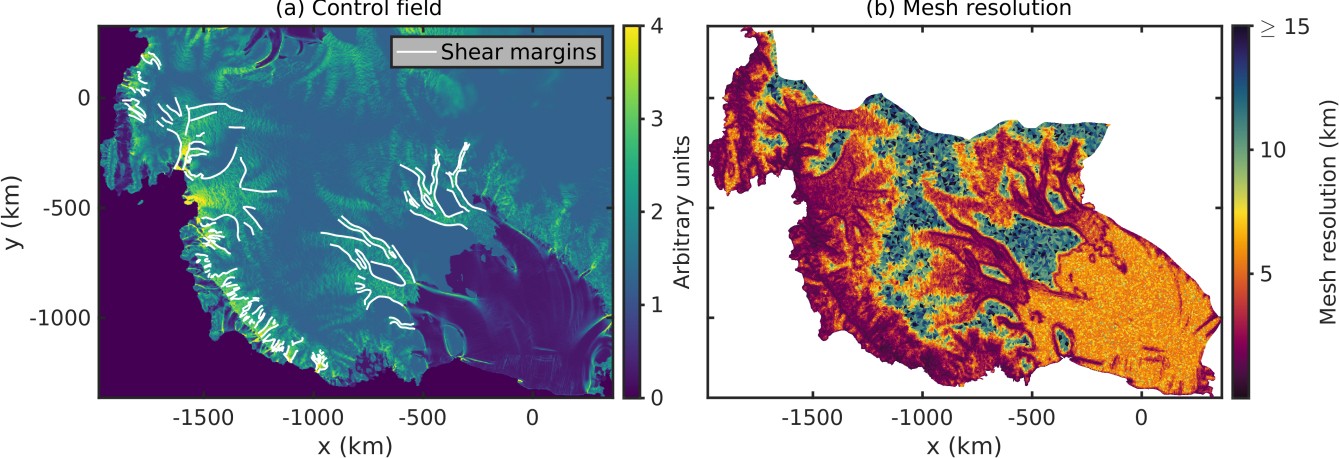

**Figure 3.** Maps of mesh characteristics. (a) The generated control field to construct the mesh with different refinements. The white colored lines indicate the detected shear margins, which are also used to create the mesh through refinement at this position (b) The resolution of the mesh with different element sizes ranging from $500\,\mathrm{m}$ to $\sim 19\,\mathrm{km}$.

The overall procedure to generate the mesh is based on a control field construction as described in Wolovick et al. (2023). In
Fig. 3 the underlying control field for the horizontal mesh generation and the element sizes for the horizontal mesh are shown. The control field (Fig. 3a) indicates the basis of the refinement strategy. For example, we want to achieve a high resolution where the ice flow is fast, demonstrated by the yellow colors in Fig. 3a, such as at the grounding line. The mesh resolution (Fig. 3b) is relatively low in the slow-flowing regions with around $15\,\mathrm{km}$ resolution and the largest elements have a resolution of $20\,\mathrm{km}$. The red and orange colors indicate a high resolution between $500\,\mathrm{m}$ and $1\,\mathrm{km}$ exactly where the grounding line and
the shear margins are located. This resolution is about a factor $3-6$ higher than in Morlighem et al. (2013) (they use $3\,\mathrm{km}$) and a factor $5-10$ higher than in Arthern et al. (2015) (they use $5\,\mathrm{km}$ for the whole domain of Antarctica). Overall the resulting mesh consists of $417,284$ 2D-elements. At the end of the meshing procedure all relevant gridded data like the bed topography (Fig. 2a), the ice thickness (Fig. 2c), as well as the observed surface velocities (Fig. 2e) and the mask (Fig. 2f) are interpolated with a multi-wavelength interpolation introduced in Wolovick et al. (2023) onto the mesh.

## 2.2 Forward model

We use a forward model that models the ice dynamics in an approximated way and is explained by the following higher-order Blatter-Pattyn approximation (HO; Blatter, 1995; Pattyn, 2003).

$$
\begin{aligned}
&\frac{\partial}{\partial x}\left(4\eta\frac{\partial v_x}{\partial x}+2\eta\frac{\partial v_y}{\partial y}\right)+\frac{\partial}{\partial y}\left(\eta\frac{\partial v_x}{\partial y}+\eta\frac{\partial v_y}{\partial x}\right)+\frac{\partial}{\partial z}\left(\eta\frac{\partial v_x}{\partial z}\right)=\rho_\mathrm{i}g\frac{\partial h_\mathrm{s}}{\partial x}\\
&\frac{\partial}{\partial x}\left(\eta\frac{\partial v_x}{\partial y}+\eta\frac{\partial v_y}{\partial y}\right)+\frac{\partial}{\partial y}\left(4\eta\frac{\partial v_y}{\partial y}+2\eta\frac{\partial v_x}{\partial x}\right)+\frac{\partial}{\partial z}\left(\eta\frac{\partial v_y}{\partial z}\right)=\rho_\mathrm{i}g\frac{\partial h_\mathrm{s}}{\partial y}
\end{aligned}
\tag{1}
$$



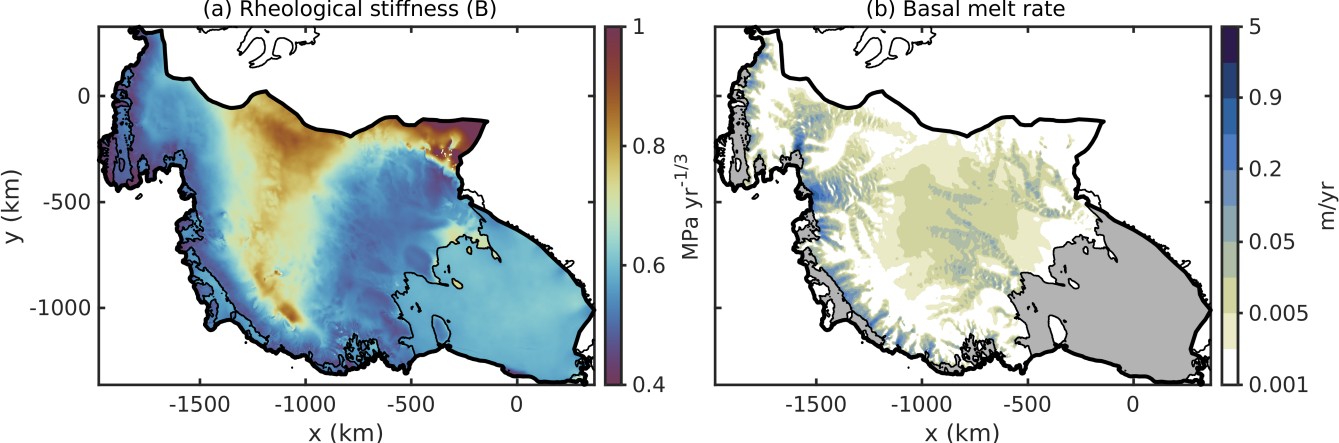

**Figure 4.** Results of the 1D steady-state advection-diffusion thermal model. (a) Vertically averaged ice rheology parameter $B$. (b) Basal melt rate of grounded ice represented with a pseudo-log scale, with gray colors representing the floating ice. The black line denotes the study area and the thinner black line represents the grounding line.

The density of ice is denoted by $\rho_i$, $g$ represents the acceleration of gravity, $h_s$ is the surface elevation of the glacier, and $\eta$ is the ice viscosity. The latter is described by the constitutive material law for ice *Glen's flow law* (Glen, 1953).

$$\eta = \frac{B}{2\dot{\varepsilon}_e^{\frac{n-1}{n}}}, \tag{2}$$

where $B$ is the ice rheology parameter, $\dot{\varepsilon}_e = \sqrt{\frac{1}{2}\text{tr}(\dot{\epsilon}_{ij}^2)}$ the effective strain-rate with $\dot{\epsilon}_{ij}$ the strain-rate tensor and $n = 3$ the flow exponent. The temperature is computed by using a 1D vertical steady-state advection-diffusion thermal model as described in Wolovick et al. (2023). To force this model, we use surface temperatures (Comiso, 2000), accumulation rates (mean of Van De Berg et al. (2005) and Arthern et al. (2006)), as well as a geothermal heat flow (sum of Martos et al. (2017) and calculated shear heating). Subsequently, we compute the flow rate factor $B$ (Cuffey and Paterson, 2010). The vertically resolved rheology $B$ is required for our HO model while the resulting basal melt rates of the thermal model (Fig. 4b) serve as an input for CUAS-MPI (compare Sect. Subglacial hydrology). Figure 4a reveals a belt in the center of the study area that exhibits increased ice stiffness associated with low surface temperatures (not shown here) and high surface elevations (compare Fig. 2b). In contrast, we can observe relatively soft ice at the margins of the domain, where warm surface temperatures and lower surface elevations predominate. The thermal model suggests that about 30% of the rectangular CUAS-MPI domain enclosing the study area in Fig. 4b are frozen to the bed. High melt rates ($> 0.5\,\text{m yr}^{-1}$w.e.) are located close to the grounding line of Pine Island and Thwaites Glacier, but those areas cover only about 0.2% of the total warm-based area. Most of the melt rates are in the mm-range (median: $\sim 6\,\text{mm yr}^{-1}$w.e.) and clearly show the fast-flowing areas.

Boundary conditions are given at the ice-atmosphere $\Gamma_s$, ice-bed $\Gamma_b$ and ice-margin $\Gamma_-$ interfaces to constrain the forward model. Towards the ice-atmosphere $\Gamma_s$ a traction-free homogeneous Neumann boundary is assumed. The lateral ice-margin boundary $\Gamma_-$ is constrained by an in-homogeneous Dirichlet condition in terms of observed surface velocities $\boldsymbol{v} = \boldsymbol{v}_s^{\text{obs}}$. At the



ice-bed interface $\Gamma_{\mathrm{b}}$ we employ a friction law acting beneath the ice sheet. This friction law includes the unknown basal drag coefficient $k$, which we want to determine with our inversion approach. In ISSM the basal friction law is implemented in terms

of basal drag $\boldsymbol{\tau}_{\mathrm{b}}$ as

$$\boldsymbol{\tau}_{\mathrm{b}} = -k^2 N^r ||\boldsymbol{v}_{\mathrm{b}}||_2^{\frac{1-m}{m}} \boldsymbol{v}_{\mathrm{b}}. \tag{3}$$

Here, $\boldsymbol{v}_{\mathrm{b}}$ is the basal velocity, $m$ the friction law exponent, $r$ the effective pressure exponent and $N = p_{\mathrm{i}} - p_{\mathrm{w}}$ denotes the effective pressure as a function of the ice pressure $p_{\mathrm{i}}$ and the water pressure $p_{\mathrm{w}}$. If the exponent of the effective pressure is $r = 0$, i.e., $N$ is neglected in Eq. (3), we obtain a Weertman-type friction law (Weertman, 1957). Otherwise, when $r > 0$ holds,

we obtain the Budd-type friction law. In the case of $m = 1$, the law is linear in terms of velocity, but in the case of $m > 1$ it becomes non-linear in velocity. Throughout the study, we set $r = 0$ when a Weertman sliding law is used and denote the corresponding basal drag coefficient $k_{\mathrm{W}}$. We set $r = 1$ when a Budd-type sliding law is employed and denote the related basal drag coefficient $k_{\mathrm{B}}$. We carry out experiments with these sliding laws using linear sliding, i.e., $m = 1$, and non-linear sliding, i.e., $m = 3$. We compare a parameterized effective pressure $N := N_{\mathrm{op}}$ against one from a hydrology model $N := N_{\mathrm{CUAS}}$,

described in Sect. 2.2 Subglacial hydrology. The effective pressure $N_{\mathrm{op}}$ is described by $N_{\mathrm{op}} = p_{\mathrm{i}} - p_{\mathrm{w}} = \rho_{\mathrm{i}} g H - \rho_{\mathrm{w}} g(-h_{\mathrm{b}})$ assuming a perfect hydrological connection to the ocean, depending on geometry data like the ice thickness $H$, as well as the bed topography $h_{\mathrm{b}}$ and the water density $\rho_{\mathrm{w}}$. Note, that we allow a negative water pressure. Details of the effective pressure $N$ are described in the next section.

**Subglacial hydrology**

In order to overcome shortcomings in the effective pressure parametrizations, we leverage a simulated effective pressure $N_{\mathrm{CUAS}}$ from the MPI parallel version of the Confined-Unconfined Aquifer System model (CUAS-MPI; Beyer et al. (2018); Fischler et al. (2023)). The model is based on an effective porous media approach (single-layer, Darcy-type flow) and solves an evolution equation for the hydraulic head

$$h = \frac{p_{\mathrm{w}}}{\rho_{\mathrm{w}} g} + h_{\mathrm{b}} + z_{\mathrm{w}}, \tag{4}$$

where $\rho_{\mathrm{w}}$ is the density of water and $0 \leq z_{\mathrm{w}} \leq b$ is the elevation within the aquifer of thickness $b$. The hydraulic transmissivity is spatially and temporally varying and evolves due to channel wall melt, creep-closure and cavity opening. This makes it possible to simulate both inefficient and efficient water transport without resolving individual channels. The ice sheet geometry for the hydrology model is also based on the BedMachine v2 dataset, but cropped out for the area shown in Fig 2 and interpolated onto a 1 km regular grid. CUAS-MPI also needs a mask to distinguish between active points and boundary conditions. This

mask is based on the interpolated bed elevation and ice thickness datasets from BedMachine taking into account the floating condition. No-flow boundary conditions are used along the grounded part of the basin delineation (black line in Fig 2) and next to ice-free land (mostly rock outcrops). To avoid water flowing into areas that are most likely to be frozen to the bed, no-flow conditions are imposed in areas where the ice thickness is below $10\,\mathrm{m}$ or the bed elevation is $2000\,\mathrm{m}$ above sea level. At the ocean boundary, a Dirichlet condition ($h = 0\,\mathrm{m}$) is applied. We initialize the head so that the water pressure is 90% of the



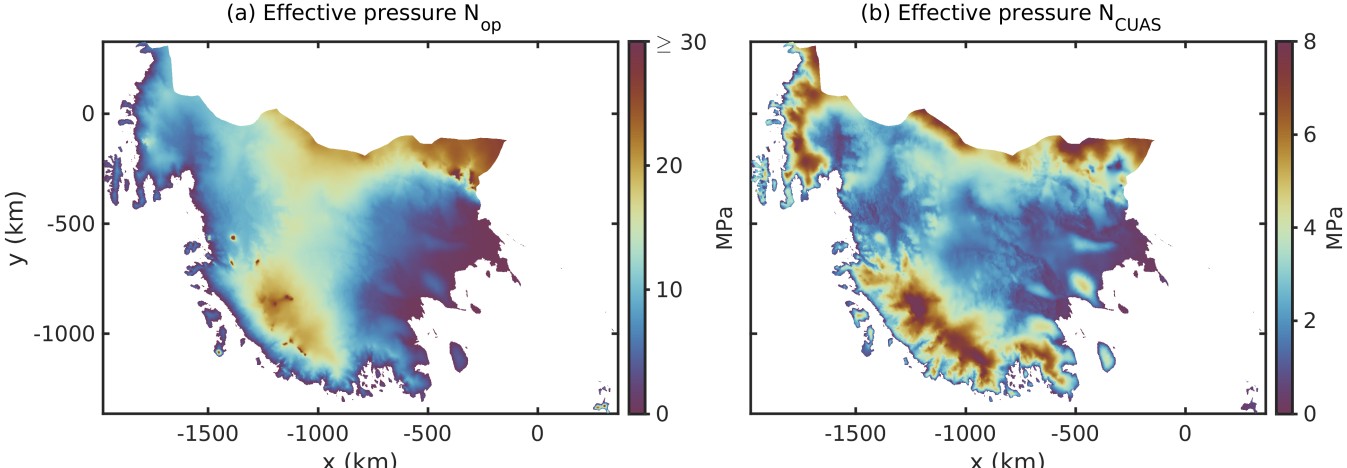

**Figure 5.** Two different effective pressure distributions for WAIS. (a) Geometry-based effective pressure $N_{op}$ ranging from $0$ to $36\,\text{MPa}$. (b) Effective pressure $N_{CUAS}$ based on the subglacial hydrology model CUAS-MPI results ranging from $0$ to $8\,\text{MPa}$, including the modifications for areas that are not part of the CUAS domain.

ice overburden pressure. The model is forced with the steady but spatially variable ice sheet basal melt distribution (Fig. 4b) based on the 1D vertical steady-state advection diffusion thermal model that is also used for rheology (Fig. 4a) in the ISSM model set-up. The basal melt rate in Fig. 4b reflects the velocity field, which is reasonable as it is included in the computation of the melt rates. The relatively high values of basal melt rate result from high rates of shear heating. We run the model with 1 hour time steps for 10 years using the same model parameters as in Beyer et al. (2018, Tab. 1 and 3) except we use an aquifer

thickness 1 m and a smaller minimum transmissivity ($T_{\min} = 10^{-14}\,\text{m}^2\,\text{s}^{-1}$).

Since we did not apply CUAS-MPI to ice rises, we set $N_{CUAS}$ to the hydrostatic pressure $N_{op}$. Also, rock outcrops shown in Fig. 2f need special treatment, as they are no data entries. We interpolate over them on the CUAS grid. Finally, we interpolate the mentioned regions to the finite element mesh. Since Thurston Island is not included in the CUAS-MPI model domain, we set it to ice pressure $p_i$ and neglect the water pressure due to the predominating low velocities at these locations.

The effective pressures $N_{op}$ and $N_{CUAS}$ are displayed in Fig. 5. In contrast to $N_{CUAS}$ (Fig. 5b), the simple parameterization of the effective pressure $N_{op}$ (Fig. 5a) reaches a much higher magnitude of $30\,\text{MPa}$ in the slow-flowing areas of the domain than the effective pressure $N_{CUAS}$ ranging only up to $8\,\text{MPa}$. Further, the structure of the effective pressure from the subglacial hydrology model $N_{CUAS}$ shows a similar distribution as the velocity field (compare Fig. 1), as expected.

## 2.3 Basal drag inversion

Parameter identification problems occur in ice sheet modeling because some relevant parameters are difficult to observe directly, like the distribution of the basal drag underneath the ice sheets. These problems are referred to as inverse problems in the



sense that we want to infer from an observed effect of a system the underlying but not measurable cause. In ice sheet models this can be represented through the observed ice surface velocity as the observed effect and the unknown basal drag coefficient.

In glaciology, such problems can be described by minimizing a cost function (Eq. (5)), while satisfying the underlying forward model and controlling the unknown basal drag coefficient $k$.

$$\min_{k} J_{\text{raw}}(\boldsymbol{v}_{\text{s}}, k) = \int_{\Gamma_{\text{s}}} \frac{1}{2}((v_x - v_x^{\text{obs}})^2 + (v_y - v_y^{\text{obs}})^2)\, d\Gamma_{\text{s}} + \lambda \int_{\Gamma_{\text{b}}} \frac{1}{2}||\nabla k||_2^2\, d\Gamma_{\text{b}}. \tag{5}$$

The first term of Eq. (5) describes the absolute velocity misfit with $\boldsymbol{v}_{\text{s}}^{\text{obs}} = (v_x^{\text{obs}}, v_y^{\text{obs}})$ the observed ice surface velocity, $\boldsymbol{v}_{\text{s}} = (v_x, v_y)$ the modeled ice velocity and $k$ the respective control parameter for the inversion. Due to the general ill-posedness of inverse problems (Hadamard, 1902), it is difficult to solve these problems. Even small measurement errors in the observed surface velocities $\boldsymbol{v}_{\text{s}}^{\text{obs}}$ can lead to significant and unrealistic artefacts in the unknown basal drag coefficient $k$ in the simulated fields. To improve this instability of the problem, it is beneficial to regularize it. Therefore the second summand is added to the cost function (Eq. (5)), describing a typical Tikhonov regularization term (Tikhonov and Arsenin, 1977) by penalizing oscillations in the basal drag coefficient $k$. This additional term is equipped with a weight $\lambda$ which can be derived by performing an L-curve analysis. A L-curve analysis is meant to be a trade-off curve between the first cost function term $J_{\text{raw,obs}}$ and the regularization term $J_{\text{raw,reg}}$.

The idea behind conducting an L-curve analysis is to pick the best weight $\lambda$ in the corner of the "L-"shape. To avoid the arbitrary choice of finding the best $\lambda$ value by hand-picking, we determine a smooth L-curve and calculate the maximum curvature for determining the best $\lambda$ value following the method of Wolovick et al. (2023). For the L-curve analysis, we use a range of $\lambda \in \left[10^{-2}, 10^4\right]$, as well as 25 logarithmically spaced samples. For every sample, a basal drag inversion is performed. We also considered other $\lambda$ ranges that included even smaller $\lambda$ values, e.g., $\lambda < 10^{-2}$. Here, using $\lambda \in \left[10^{-3}, 10^3\right]$ (compare Sect. 3), we obtain an extended small-$\lambda$ limb for the inversion runs including non-linear friction laws and can thus determine the corner of the L-curve more clearly. At very small $\lambda$ values we observed that no convergence is achieved for linear friction laws. From a mathematical point of view, this is reasonable, as the problem is again non-convex due to the non-linearity of the forward model (Eq. (1)) and the regularization contributes too little. In addition, the initial guess of the basal drag coefficient $k$ also has an influence on the convergence of the inversion(compare convergence criteria Eq. (9)). Therefore, it must be well quantified. A good approximation for the initial basal drag coefficient $k_{\text{init}}$ can be computed from the driving stress and the observed ice velocity as (compare, e.g., Morlighem et al. (2013))

$$k_{\text{init}} = \left( \frac{\max(0, \tau_{\text{d}})}{N||\boldsymbol{v}_{\text{s}}^{\text{obs}}||_2^{1/m}} \right)^{1/2}, \tag{6}$$

where $\tau_{\text{d}} = -\rho_{\text{i}} g H \nabla h_{\text{s}} \cdot \boldsymbol{v}_{\text{s}}^{\text{obs}}/||\boldsymbol{v}_{\text{s}}^{\text{obs}}||_2$ the driving stress in flow direction. In the case of Weertman friction, $N$ is set to 1 in Eq. (6) and for the Budd-type friction law a minimum value of $100\,\text{Pa}$ is used. For the velocity we use a minimum value of $0.1\,\text{m}\,\text{yr}^{-1}$ to also prevent division by zero.

We observed that it can be useful to smooth the initial value $k_{\text{init}}$ slightly, especially in the case of the non-linear friction law with $m = 3$, in order to reach convergence. The reason for this is probably that very fine structures in the initial basal drag





coefficient $k_{\text{init}}$ are too complex for the solver, as the convergence criterion depends on the initial drag coefficient $k_{\text{init}}$ (Eq. (9))
and also it may not be possible to find a solution for the stress balance in this case.

Following again Wolovick et al. (2023), we introduce a characteristic scale to normalize our cost function in Eq. (5) before we carry out the inversion procedure as described above. This is important to ensure that the parameter space (range of $\lambda$ values) analyzed in the L-curve is easy to identify and interpret. This is the case, when $\lambda$ values are unitless and of order unity. Thus, the optimal regularization weight $\lambda$ in the maximum curvature can be found more easily in the parameter space and it is 265 easier to evaluate whether the degree of regularization is small or large. The terms of the scaled cost function $J(\boldsymbol{v}_{\text{s}}, k)$ can be described as

$$J(\boldsymbol{v}_{\text{s}}, k) = J_{\text{obs}}(\boldsymbol{v}_{\text{s}}) + \lambda J_{\text{reg}}(k), \quad \text{with}$$

$$J_{\text{obs}}(\boldsymbol{v}_{\text{s}}) = \frac{1}{S_{\text{obs}}} \int\limits_{\Gamma_{\text{s}}} \frac{1}{2}((v_x - v_x^{\text{obs}})^2 + (v_y - v_y^{\text{obs}})^2)\, d\Gamma_{\text{s}} \quad \text{and} \tag{7}$$

$$J_{\text{reg}}(k) = \frac{\lambda}{S_{\text{reg}}} \int\limits_{\Gamma_{\text{b}}} \frac{1}{2}||\nabla k||_2^2\, d\Gamma_{\text{b}}.$$

The scaling terms $S_{\text{obs}}$ and $S_{\text{reg}}$, which are described through the a-priori estimation of the characteristic magnitudes of the corresponding terms, are

$$S_{\text{obs}} = \int\limits_{\Gamma_{\text{s}}} ||\boldsymbol{v}_{\text{s}}^{\text{obs}}||_2^2\, d\Gamma_{\text{s}} = A\sigma_{\text{obs}}^2 \quad \text{and}$$

$$\tag{8}$$

$$S_{\text{reg}} = A\left(\frac{\pi\sigma_k}{\bar{H}}\right)^2.$$

Here $S_{\text{obs}}$ is defined by the variance of the surface velocity observations $\boldsymbol{v}_{\text{s}}^{\text{obs}}$, where $A$ is the area of the grounded domain and $\sigma_{\text{obs}}^2$ the root-mean-squared (RMS) amplitude of the observed surface velocity. The scaled regularization term $S_{\text{reg}}$ has the same magnitude as the final drag coefficient guess, where $A$ is again the grounded domain area, $\sigma_k$ the standard deviation of $k_{\text{init}}$ (Eq. (6)) and $\bar{H}$ the mean ice thickness. A more detailed description of the derivation of $S_{\text{reg}}$ can be found in Wolovick et al. 275 (2023).

Our non-linear forward model (Eq. (1)) is solved with a Picard iteration scheme (iterative fixed-point method, Hindmarsh and Payne (1996); Smedt et al. (2010)) implemented in ISSM. The remaining linear systems of equations are solved with an iterative GMRES (Generalized Minimal Residual Algorithm, Saad and Schultz (1986)) solver combined with a Block Jacobi preconditioner provided by PETSc (Balay et al., 1997, 2019). For simplification, the viscosity is set independent of the velocity 280 in the adjoint equations (e.g., MacAyeal, 1992; Morlighem et al., 2013). The linear adjoint equations obtained, represent a good approximation to the exact adjoint equations (Morlighem et al., 2013). In order to minimize the cost function of our basal drag inversion a limited quasi-Newton technique the Limited-memory Broyden-Fletcher-Goldfarb-Shanno algorithm (L-BFGS, Nocedal (1980)) called M1QN3 (Gilbert and Lemaréchal, 1989) is used. This algorithm provides two different convergence criteria, the cost function convergence criterion $\Delta_{x_{\text{min}}}$ and the gradient relative convergence criterion $\epsilon_{\text{gttol}}$, described in Eq. (9).






$$||J(v_i, k_i) - J(v_{i+1}, k_{i+1})|| < \Delta_{x_{\min}}$$
$$\frac{||\nabla J(v_i, k_i)||}{||\nabla J(v_0, k_0)||} < \epsilon_{\text{gttol}} \tag{9}$$

Where $\nabla J$ denotes the gradient of the cost function $J$ at $(v_i, k_i)$ with $i$ displaying the iteration steps. The initial guess is described through $v_0 := \boldsymbol{v}_{\text{s}}^{\text{obs}}$ and $k_0 := k_{\text{init}}$.

# 3 Results

First, we analyze the corresponding L-curves of the six experiments (two types of friction laws and three effective pressure realizations), as well as the optimization convergence behavior. We present a subdomain L-curve analysis for eight different subdomains of our study area in order to explore the dependence of regularization and inverse problem ill-posedness on glaciological settings. We examine the influence of the effective pressure realizations and the linear and non-linear friction laws on the inferred basal drag distributions, as well as on the L-curves. Subsequently, we show the spatial distribution of the 295 best estimate basal drag of our study area.

## 3.1 L-curve analysis

In Fig. 6 the L-curves for all six conducted experiments are shown. The first row of Fig. 6 displays the L-curves with linear sliding $m = 1$ for Weertman and Budd with $N_{\text{op}}$ and $N_{\text{CUAS}}$; the second row shows the L-curves with non-linear sliding $m = 3$, respectively. Each subplot (Fig. 6a–f) illustrates the data costs $J_{\text{obs}}$ in relation to the regularization costs $J_{\text{reg}}$ with the black 300 dots representing an inversion run for each of the 25 different $\lambda$ values. The smoothed trade-off curve is characterized by the black line which lies through the 25 different model points. The red line indicates in which sector of each L-curve a $\lambda$ value should be chosen. This corner region ranges from the maximum $\lambda_{\max}$ value to the minimum $\lambda_{\min}$ value. Where $\lambda_{\text{best}}$, displayed in Fig. 6, describes the maximum curvature of the respective curve.

Note that the L-curves with a linear friction law $m = 1$ are given for a range of $\lambda \in [10^{-2}, 10^4]$ and those using a non-linear 305 friction law for a range of $\lambda \in [10^{-3}, 10^3]$ (compare Sect. 2.3). This is because the resulting L-curves with linear sliding are very flat when using a lower range of $\lambda \in [10^{-3}, 10^3]$. This makes it even more difficult to select a corner region, especially when the vertical limb is barely recognizable. Conversely, a steeper form of the "L" can be recognized for the L-curves with non-linear friction laws and a higher range of $\lambda \in [10^{-2}, 10^4]$. In this case, the flat limb for $m = 3$ is not prominent enough, which in turn makes it again difficult to select a corner region and thus the $\lambda_{\text{best}}$ value. To obtain a proper L-curve shape in 310 which both, the corner region and the $\lambda_{\text{best}}$ value can be optimally determined, we decided to use the most suitable ranges for the respective L-curves. In addition, the characteristic scale (Eq. (8)), which we introduced in Sect. 2.3, helps us to find a matching $\lambda$ range that contains both limbs of the L-curve and thus guarantees a valid corner region.

We also found that if the $\lambda$ range for $m = 1$ is taken too low, many outliers can be found among the smaller $\lambda$ values. We already mentioned in Sect. 2.3 that we observed difficulties in achieving convergence for $\lambda$ values that are too small, especially







**Figure 6.** All six conducted L-curves in a log-log plot for the regularization cost $J_{\mathrm{reg}}$ against the data cost $J_{\mathrm{obs}}$. The black dots indicate the inversion result for each $\lambda$ value, the black line describes the smooth trade-off curve and the red color shows the corner region of the L-curves. Gray dots indicate outliers. (a) L-curve result for the linear Weertman friction law with $\lambda_{\mathrm{best}} = 0.33$ (b) L-curve for the linear Budd-type friction law including effective pressure from geometry $N_{\mathrm{op}}$ with $\lambda_{\mathrm{best}} = 2.4$ (c) L-curve result for the linear Budd-type friction law using the effective pressure from CUAS-MPI $N_{\mathrm{CUAS}}$ with $\lambda_{\mathrm{best}} = 1.3$ (d) L-curve for the non-linear Weertman friction law with $\lambda_{\mathrm{best}} = 2.4$ (e) L-curve result including linear Budd-type friction law for the effective pressure from geometry $N_{\mathrm{op}}$ with $\lambda_{\mathrm{best}} = 0.11$ (f) L-curve for the effective pressure $N_{\mathrm{CUAS}}$ with $\lambda_{\mathrm{best}} = 0.5$.

when a linear friction law is chosen for the inversion in a range of $\lambda \in \left[10^{-3}, 10^{3}\right]$. This is rather a numerical problem than one resulting from the mechanical system of the model, since the problem becomes non-convex if the regularization is too low, i.e., if the $\lambda$ values are really small. However, one possibility in this case is to avoid small $\lambda$ values, which are unimportant from a mathematical perspective anyway, by shifting the $\lambda$ range upwards to higher values. In addition, we noticed, that inversion runs





using a non-linear friction law have more convergence difficulties than runs with a linear friction law. In our case, a further
smoothing of the initial drag coefficient $k_{\text{init}}$ (see Eq. (6)) improved the convergence results. This makes the initial field simpler
and less noisy and thus easier to find a solution for the stress balance solver. In the case of inversion runs using a non-linear
friction law we had to smooth the initial drag coefficient $k_{\text{init}}$ far more than in the case of linear friction laws. In general, it
helps to further adjust the convergence criteria $\epsilon_{\text{gttol}}$ and $\Delta_{x_{\min}}$ (Eq. (9)), if the convergence of the optimization is not achieved.
For example, when an inversion only converges with $\Delta_{x_{\min}}$ during the line search, it is recommended to set $\Delta_{x_{\min}}$ more strictly.
In our case, we achieved the best results with $\epsilon_{\text{gttol}} = 10^{-3}$ and $\Delta_{x_{\min}} = 10^{-5}$ for inversion runs using a linear friction law. For
convergence of the L-curve runs with a non-linear friction law, we had to set the convergence criterion much more strictly to
$\epsilon_{\text{gttol}} = 10^{-6}$ and $\Delta_{x_{\min}} = 10^{-4}$. This was also observed in the study by Wolovick et al. (2023) when non-linear friction laws
are used. Overall, we were able to significantly reduce the outliers of the different L-curves and achieved convergence for most
of the inversion runs, when using the mentioned adjustments.

Despite the choice of a different, upward-shifted $\lambda$ interval, the L-curves with a linear friction law are still very flat in
shape compared to those of the non-linear friction law. Nevertheless, visually all of these six L-curves have the desired and
good-looking "L"-shape. This impression can be trusted, because we use a log-log plot with the same scaling for both axes as
described in Wolovick et al. (2023), e.g., Fig. 3. Overall, we can observe in each of the six L-curves a smooth monotonic result
with a maximum of one outlier per L-curve. In addition, the smooth trade-off curve of each L-curve fits the model points very
well. The $\lambda_{\text{best}}$ value, which we pick by the maximum curvature of our curve fitting procedure is relatively in the corner of
almost all of the L-curves. Except for the $\lambda_{\text{best}}$ value in Fig. 6a,e, which would probably be selected visually at a slightly higher
value. This is due to the fact, that we pick the peak curvature to choose the $\lambda_{\text{best}}$ value. Looking at Fig. 9b–d and Fig. 10b–d,
which represents the corresponding cost curvature, we observe that every shown total cost curvature in black has a clear peak
except the one of Fig. 9c and Fig. 10c corresponding to the experiments in Fig. 6a,e. In these two figures the total curvature
has a broad region of generally high curvature representing the visual corner and with a narrow small peak at the side. In our
method, this side peak is selected instead of the mean value of this broader region, which explains why the $\lambda_{\text{best}}$ is not selected
exactly in the corner of the L-curve.

For all six L-curves together, the $\lambda_{\text{best}}$ value ranges between $[0.11, 2.4]$. Compared to other L-curve results we obtained, not
shown in this manuscript, we can conclude that this is a relatively small range of $\lambda$ values. Overall, these good results from the
L-curves of the six experiments give us confidence that our model procedure is trustworthy.

Figure 7 shows the convergence of the total cost function $J$ for all six experiments of the optimization based on the $\lambda_{\text{best}}$
weight (labeled in Fig. 6). The $x$-axis displays the number of iterations required for the cost function $J$ to reach an optimum.
In the case of a linear friction law (Fig. 7a), 200 iterations are required on average. However, if a non-linear friction law is
used (Fig. 7b) fewer iteration steps of around 120 are required until convergence is reached. But it is remarkable that the run
with a non-linear Budd friction law employing $N_{\text{op}}$ requires up to 60 iterations more than the other two runs using a non-linear
friction law. In the case of the linear friction law (Fig. 7a), the inversion using a Weertman friction law requires significantly
more iterations (about 70) for convergence than the two with the Budd-type friction law. In both, the linear and non-linear cases,



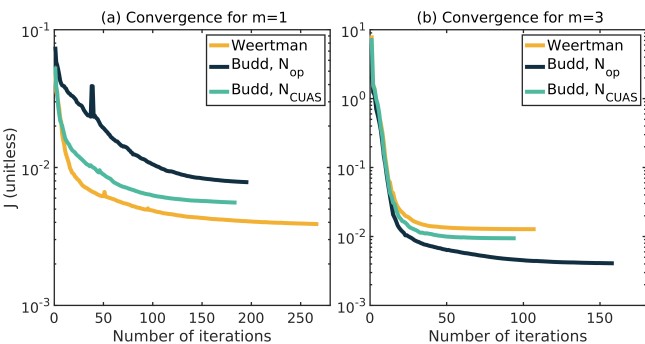

**Figure 7.** Convergence of cost function $J$ for the respective best weight $\lambda_{\text{best}}$ of all six optimization runs. Note the different ranges of iteration steps for the $x$-axis. (a) Convergence for the three experiments using a linear friction law. (b) Convergence for the three experiments with a non-linear friction law. The yellow line indicates the convergence of the inversion using a Weertman friction law, the dark blue line denotes the convergence of the inversion run with Budd-type and $N_{\text{op}}$ and the turquoise color characterizes the convergence of the run with Budd-type using $N_{\text{CUAS}}$, respectively for plot (a) and plot (b).

the inversions involving a Budd friction law with the effective pressure $N_{\text{CUAS}}$ require the least iterations until convergence is reached.

**Subdomain L-curve analysis**

In order to explore how the optimal regularization and the ill-posedness of the inverse problem depends on glaciological settings, we analyze L-curves for selected subsets of our domain. The subdomain L-curve analysis is performed for a linear Budd-type friction law experiment marked with blue color (Fig. 8) as well as for the non-linear Budd friction law represented in red color (Fig. 8) including $N_{\text{CUAS}}$. Figure 8a shows the eight different subdomains, with the goal of including a wide variety

of glaciological settings. We choose subdomains representing the main trunks of Pine Island (Fig. 8b) and Thwaites Glaciers (Fig. 8c), upstream tributaries of Thwaites (Fig. 8d), a slow-flowing region around WAIS Divide (Fig. 8e), tributary glaciers of Whillans that cross the Trans-Antarctic Mountains in the presence of many rock outcrops (Fig. 8f), the shear margin of Whillans Ice Stream (Fig. 8g), the fast-flowing trunks of Bindschadler and MacAyeal Ice Streams (Fig. 8h), and finally one subdomain including Roosevelt Island (Fig. 8i).

For each subdomain, we compute the cost function components $J_{\text{obs}}$ and $J_{\text{reg}}$ including only model elements within the subdomain region. Since we do not recalculate the characteristic scaling factors for these subdomains or adjust for the fact that we are integrating over a smaller area, these subdomain L-curves are shifted towards smaller values relative to the L-curve for the entire domain, but the relative shape of the subdomain L-curves should still reflect the glaciological differences between these regions.

The subdomain analysis results in a variety of different L-curve behaviors for both shown experiments (Fig. 8). When assuming a linear friction law, Figure 8 demonstrates in blue color that the L-curves of the various subdomains provide very smooth results and outliers are only detected for very small $\lambda$. Especially the subdomains Fig. 8b,c, as well as f–h show a





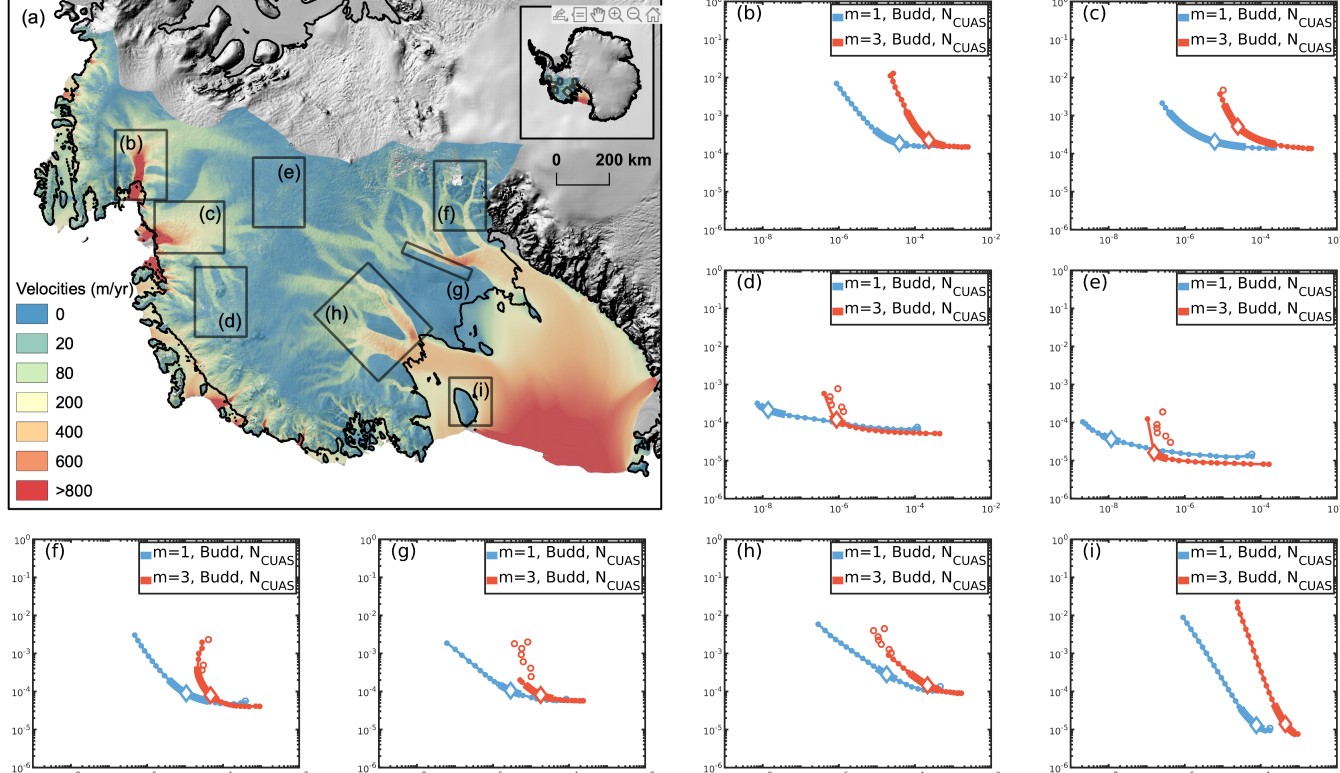

**Figure 8.** Subdomain L-curve using the data cost $J_{\mathrm{obs}}$ ($y$-axis) and regularization cost $J_{\mathrm{reg}}$ ($x$-axis) from the experiments $m = 1$, $N_{\mathrm{CUAS}}$ and $m = 3$, $N_{\mathrm{CUAS}}$. (a) Shows the study area plot from Fig. 1 with the eight selected subdomains marked by the dashed black lines. The labels (b–i) of the different subdomains reflect the corresponding L-curves. (b–i) display the eight L-curves associated to the subdomains in blue color for the experiment $m = 1$, $N_{\mathrm{CUAS}}$ and in red color for the experiment $m = 3$, $N_{\mathrm{CUAS}}$.

well-behaved L-curve with a good curvature. In contrast, when assuming a non-linear friction law, we obtain the smoothest L-curve (red color) for the subdomain including the Pine Island Glacier (Fig. 8b–i). Furthermore, the subdomain including

the rock outcrops region (Fig. 8f) and the subdomain covering Thwaites Glacier (Fig. 8c) reveal relatively smooth L-curves with a well-defined curvature. But overall, we get many outliers in the L-curves for larger $\lambda$ using the non-linear friction law such as in the subdomains of the WAIS divide (Fig. 8b), in the area of shear margins from Whillans Ice Stream (Fig. 8h), in the upstream tributary of Thwaites Glacier (Fig. 8c), as well as in the L-curve of the MacAyeal and Bindschadler Ice Streams (Fig. 8h). Showing us, that the smooth L-curve incorporating the whole model domain (Fig. 6f) suppresses those outliers for

increased $\lambda$ values. But as mentioned previously in Sect. 3.1 we observed the effect that when using non-linear sliding many outliers appear for higher $\lambda$ values in the L-curves if the lambda range is shifted towards higher $\lambda$ values. Therefore we shifted the $\lambda$ range further downwards. This shift for $\lambda$ could possibly already lead to an improvement in the smoothness of the L-curve when integrating over the entire domain, but not for the individual subdomains. This could also be explained by the fact that some of the subdomain L-curves are very smooth and have fewer outliers, and if these predominate, the few outliers in other





areas are suppressed during integration over the entire domain. The subdomain covering Roosevelt Island generated a quite steep L-curve for both linear and non-linear sliding, with only a slight hint of a corner at the lower end (e.g., Fig. 8i). It is possible that this region might show a stronger corner in its L-curve if we extended the analysis to smaller $\lambda$ values. However, observed velocity is very slow in this ice rise, and the apparent flattening at the low end may simply reflect the inability of the inverse model to reduce misfit below the noise level of the observations. In that case, the straight-line (power-law) shape

of the Roosevelt Island L-curve may reflect the fact that this ice rise is likely frozen to the bed (Martín et al., 2006), and thus is poorly suited to a basal sliding inversion. In contrast, we observe in the results for both L-curves in Fig. 8 that the subdomain in the upstream tributary of Thwaites (Fig. 8c) as well as the subdomain in the WAIS divide (Fig. 8e) reveal a very flat L-curve. This suggests that subdomain L-curves characterized by lower velocities have no major relevance for the individual basal drag inversions, as $J_{\mathrm{obs}}$ remains for each regularization weight $\lambda$ in a similar order of magnitude. Indicating

that not much regularization is required, as the variability in the basal drag coefficient might be lower in these regions. This is especially the case when linear sliding is used. It could be possible, that a shift towards a higher $\lambda$ range would show a proper curve. For $m = 3$ this is not the case, as we have significantly more outliers in both regions for higher $\lambda$ values, as mentioned before. Conspicuous is that the subdomain covering many rock outcrops of the tributary glaciers of Whillans (Fig. 8f) shows a relatively smooth behavior for $m = 1$ and $m = 3$. Since the modeling of rock outcrops is not straightforward, this gives us

confidence in our treatment of modeling those areas.

      If we compare the L-curves of the subdomains in Fig 8 for the experiment $m = 3, N_{\mathrm{CUAS}}$) to those of the experiment $m = 1, N_{\mathrm{CUAS}}$, we can recognize that there are significantly fewer outliers for the linear sliding case, which only occur for very small $\lambda$ and the L-curves also appear smoother. In general, this is consistent with the results obtained for the L-curves of the entire domain, because here the L-curves using linear sliding (Fig. 6a,c) also exhibit more outliers for smaller $\lambda$ and

the L-curves with non-linear sliding law (Fig. 6d–f) displays only outliers for larger $\lambda$. Despite this different behavior, the general shape in terms of steepness and flatness of the L-curves remains the same for non-linear and linear sliding across all subdomains. But as observed in Sect. 3.1, the L-curves including non-linear sliding exhibit a steeper behavior.

      We can conclude that when using linear sliding laws, the different glaciology settings have no recognizable impact on the L-curves, since only a few outliers occur (Fig. 8). However, if a non-linear sliding law is used, the influence of different glacio-

logical factors seem to be of greater importance. We therefore suggest to perform a subdomain L-curve analysis, whenever a basal drag inversion for a larger model domain is considered, such as WAIS. In particular, when those inversions lead to unexplainable issues, this analysis can be used to find errors that are not directly accessible or even to discover specific regions that could cause these problems. Overall, we can learn from this subdomain L-curve analysis how the L-curve behaves for regions with different glaciological settings. We observe that the region covering shear margins causes the biggest problem when

a non-linear sliding law is included. From slow-flowing areas like the subdomain of WAIS divide or the upstream tributary of Thwaites Glacier we conclude that they have no significant influence on the regularization regardless of whether linear or non-linear sliding is used. The areas of rock outcrops have not such a big influence on the L-curve as we expected. Considering the subdomains that include fast-flow regions like the Thwaites Glacier, the Pine Island Glacier or the Siple Coast for both linear and non-linear sliding, we observe a relatively smooth behavior of the subdomain L-curves.



**Table 1.** Table of statistical values for the grounded domain (Fig. 2f) of all six conducted experiments each evaluated at its $\lambda_{\text{best}}$ point. The reference experiment with the Weertman sliding is denoted by $k_{\text{W}}^2$ and $J_{\text{obs,W}}$ with $m=1$ and $m=3$ respective, depending on which experiment is considered. We consider the variance $\sigma^2$ of the logarithmic squared basal drag coefficient $\ln(k^2)$ and the logarithmic basal drag $\ln(\tau_{\text{b}})$, as well as the variance ratio of both. The correlation $R^2$ of effective pressure $N$ and basal drag coefficient $k^2$ regarding the reference experiment $k_{\text{W}}^2$ is taken into account. The table displays the observational costs $J_{\text{obs}}$ and the velocity equivalent of $J_{\text{obs}}$ scaled by the velocity absolute scale RMS. For the valuation of the whole model, we inspect the total model ratio with respect to the variance $\sigma^2$ of the ratio $\ln(k^2)$ to $\ln(k_{\text{W}}^2)$ times the ratio of velocity misfit $J_{\text{obs}}$. The Table 1 from Wolovick et al. (2023) serves as a reference for this table.

| Experiments (grounded domain) | $k^2$ var $\sigma^2(\ln(k^2))$ (unitless) | $\tau_{\text{b}}$ var $\sigma^2(\ln(\tau_{\text{b}}))$ (unitless) | Var ratio $\frac{\sigma^2(\ln(k^2))}{\sigma^2(\ln(\tau_{\text{b}}))}$ (unitless) | Correlation $R^2(N,k_{\text{W}}^2)$ (unitless) | Obs cost $J_{\text{obs}}$ ($\times 10^{-4}$) | Equiv $\Delta\boldsymbol{v}_{\text{s}}$ RMS$\times\sqrt{J_{\text{obs}}}$ (m yr$^{-1}$) | Total var ratio $\frac{\sigma^2(\ln(k^2))}{\sigma^2(\ln(k_{\text{W}}^2))}\times\frac{J_{\text{obs}}}{J_{\text{obs,W}}}$ (unitless) |
|---|---|---|---|---|---|---|---|
| $m=1$, Weertman | 9.07 | 4.14 | 2.19 | - | 38.49 | 20.48 | 1.00 |
| $m=1$, Budd, $N_{\text{op}}$ | 4.86 | 2.54 | 1.91 | 0.28 | 58.55 | 25.26 | 0.81 |
| $m=1$, Budd, $N_{\text{CUAS}}$ | 6.28 | 2.78 | 2.26 | 0.38 | 46.96 | 22.62 | 0.84 |
| $m=3$, Weertman | 1.91 | 1.29 | 1.48 | - | 86.46 | 30.70 | 0.47 |
| $m=3$, Budd, $N_{\text{op}}$ | 3.96 | 3.09 | 1.28 | 0.28 | 39.90 | 20.85 | 0.45 |
| $m=3$, Budd, $N_{\text{CUAS}}$ | 1.72 | 1.77 | 0.97 | 0.38 | 71.84 | 27.98 | 0.35 |

## 3.2 Effective pressure

In order to test the influence of subglacial hydrology realizations on the basal drag inversion results, we employ friction laws both with (Budd) and without (Weertman) effective pressure included. For the Budd-type law, we test both an effective pressure $N_{\text{op}}$ obtained from geometry, Fig. 5a), as well as one taken from the hydrology model CUAS-MPI $N_{\text{CUAS}}$ (Fig. 5b). We first analyze the impact of using effective pressure on our six L-curves (Fig. 6) followed by the effect on the spatial distribution of the basal drag.

Figure 9 a displays the three different L-curves that we obtain when using a non-linear friction law $m=3$. The dark blue colored L-curve including $N_{\text{op}}$ in the friction law is very close to the L-curve using a Weertman-type friction law regarding the $J_{\text{obs}}$ and $J_{\text{reg}}$ values. In contrast, the turquoise L-curve including $N_{\text{CUAS}}$ in the friction law is a bit steeper and shifted towards higher $J_{\text{reg}}$ values. In addition, the L-curve is moved further upwards to a higher $J_{\text{obs}}$ error. This is supported by a value of $J_{\text{obs}} = 71.84\,\text{m yr}^{-1}$ for $m=3$ and $N_{\text{CUAS}}$ which is relatively high compared to $J_{\text{obs}} = 39.90\,\text{m yr}^{-1}$ for $m=3$ and $N_{\text{op}}$ (Table 1). But overall, we experienced fewer convergence problems with using $N_{\text{CUAS}}$ in the Budd friction law for the L-curve illustrated with turquoise color. This is also emphasized by the absence of outliers, resulting in a very smooth L-curve. The $\lambda_{\text{best}} = 0.5$ value for the turquoise-colored L-curve is located between the other two $\lambda_{\text{best}}$ values of the experiments and, from a visual perspective, fairly matches the corner of the L-curve. In comparison, Fig. 10 shows the three L-curves for the linear friction law case. Here, the $\lambda_{\text{best}}$ value of the L-curve with $N_{\text{CUAS}}$ is again in the middle of the $\lambda_{\text{best}}$ value of





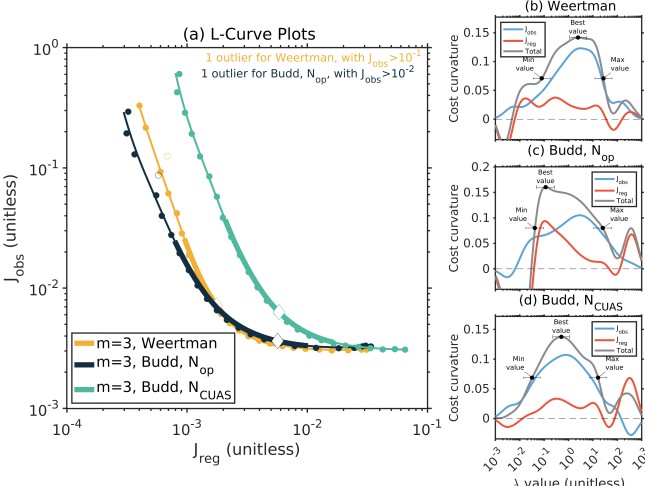

**Figure 9.** L-Curves (data cost $J_{\text{obs}}$ vs. regularization cost $J_{\text{reg}}$) with non-linear Weertman- and Budd-type friction laws. (a) L-curve for non-linear Weertman sliding in yellow color, for the non-linear Budd friction law including $N_{\text{op}}$ in dark blue, with $N_{\text{CUAS}}$ in turquoise. The solid line represents the smooth trade-off curve, the thicker line characterizes the corner region and the white diamond marks the $\lambda_{\text{best}}$ value in each of these L-curves. (b–d) display the cost curvature $\frac{d^2(ln(J))}{d(ln(\lambda))^2}$ dependent on $\lambda$ in the order for Weertman sliding (no $N$), Budd-type with $N_{\text{op}}$ and $N_{\text{CUAS}}$. Where the gray line represents the total cost function $J$, the red line the regularization cost function term $J_{\text{reg}}$ and the blue line the velocity misfit cost function term $J_{\text{obs}}$ in each of these three plots. One inversion run is detected as an outlier for Weertman sliding and Budd, $N_{\text{op}}$, respectively illustrated through the non-filled circle.

the two other L-curves, but now significantly higher at $\lambda_{\text{best}} = 1.3$. In addition, the L-curve using linear Weertman sliding has now a significantly lower $\lambda_{\text{best}} = 0.33$ value as in the non-linear sliding case (compare Fig. 6d). The one with $N_{\text{op}}$ is significantly higher with $\lambda_{\text{best}} = 2.4$ compared to the non-linear friction law (compare Fig. 6e). Moreover, the L-curve for $N_{\text{op}}$ and the one for $N_{\text{CUAS}}$ are almost identical in the linear case. This is again emphasized by the same order of magnitude from

$J_{\text{obs}} = 46.96\,\text{m yr}^{-1}$ for $m = 1$, $N_{\text{CUAS}}$ and $J_{\text{obs}} = 58.55\,\text{m yr}^{-1}$ for $m = 1$, $N_{\text{op}}$. The L-curve using a linear Weertman friction law shows a value of $J_{\text{obs}} = 38.49\,\text{m yr}^{-1}$ which fits with the slight upwards shift of the L-curve (Fig. 10a). In total, we get the best performance of the L-curves regarding the $J_{\text{obs}}$ and $J_{\text{reg}}$ for using Weertman sliding regardless of using linear or non-linear sliding. All $J_{\text{obs}}$ values in Table 1 (column five) match exactly with the $\lambda_{\text{best}}$ values of the L-curves for the different experiments (Fig. 6). High $\lambda_{\text{best}}$ values indicate a higher surface velocity mismatch and vice versa.

In order to compare the inferred basal drag based on Weertman sliding to the resulting basal drag distribution based on Budd-type sliding, we should again consider Eq. 3. To reproduce the same velocity and stress fields for both sliding laws, it can be assumed that the equation $k_{\text{W}}^2 = k_{\text{B}}^2 N$ must apply, where $k_{\text{W}}^2$ describes the squared basal drag coefficient according to Weertman sliding and $k_{\text{B}}^2$ according to Budd sliding. The Budd drag coefficient $k_{\text{B}}^2$ must satisfy this equation, if a stress field of Weertman sliding is assumed. From this assumption, we can conclude that we benefit from the effective pressure field $N$

if a linear correlation between $N$ and $k_{\text{W}}^2$ prevails. When considering a weak correlation between these terms, then $k_{\text{B}}^2$ must



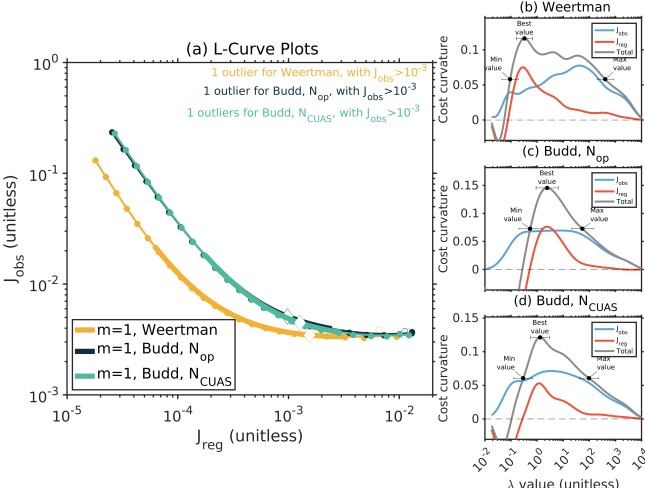

**Figure 10.** L-Curves (data cost $J_{\text{obs}}$ vs. regularization cost $J_{\text{reg}}$) with linear Weertman- and Budd-type friction laws. The description is equivalent to the one from Fig. 9, only for the linear friction law instead of the non-linear friction law. One inversion run is detected as an outlier (non-filled circle) for each of the three L-curves.

consider a large amount of structure to fit the observations. Conversely, a high linear correlation between $N$ and $k_{\text{W}}^2$ would result in a $k_{\text{B}}^2$ accounting for less structure. In the ideal limit, $N$ would represent the entire spatial structure in $k_{\text{W}}^2$ and $k_{\text{B}}^2$ would be constant. In this case, the respective resulting basal drag distribution $\tau_{\text{b}}$ should not differ significantly (compare Fig. 17a and Fig. 18a). However, this only applies if $\lambda_{\text{best}}$ is selected equally in both runs, as otherwise the resulting fields would be

smoother for one run, which in turn would lead to different basal drag distributions. In addition, both resulting cost functions $J$ should converge to the same minimum or at least have the same order of magnitude. Since a run may stop too early or not converge at all, resulting again in two different basal drag distributions. Overall, we would expect that changing the sliding law would not have a large effect on the resulting basal drag field, but the basal drag coefficient inferred from the inversion would differ significantly.

We can thus investigate the quality of both $N$ fields by computing the linear correlation between $N$ and $k_{\text{W}}^2$ (Wolovick et al., 2023). If this results in a positive correlation of $N$ and $k_{\text{W}}^2$ the basal drag inversion will not generate much structure in $k_{\text{B}}^2$. Table 1 (fourth column) displays for all of our experiments a positive correlation $R^2$ of $N$ regarding $k_{\text{W}}^2$. In total, $N_{\text{CUAS}}$ shows a slightly stronger positive correlation with $k_{\text{W}}^2$ than $N_{\text{op}}$. This result matches the distribution of $k_{\text{B}}^2$ recognizable in Fig. 11.

    Figure 11 displays the resulting drag coefficient distribution $k^2$ in a logarithmic scale for $m = 1$ in the first row and $m = 3$

in the second row with $N_{\text{op}}$ and $N_{\text{CUAS}}$ respectively. For better comparison, each basal drag coefficient is evaluated at the $\lambda_{\text{best}}$ of the respective $N_{\text{CUAS}}$ experiment. When comparing the structure of the basal drag coefficient shown in the subplots Fig. 11a,b, as well as the distribution of basal drag coefficient in the second subplot row Fig. 11c,d, we observe that for all patterns the $N_{\text{op}}$ experiments exhibits more structure in $k^2$ than the experiments with $N_{\text{CUAS}}$. This finding is emphasized by the correlation $R^2(N, k_{\text{W}}^2)$ in Table 1, as $N_{\text{CUAS}}$ has a stronger correlation with $k^2$ than $N_{\text{op}}$. Especially, when we take a closer



**Figure 11.** Drag coefficient maps of $\log10(k^2)$. (a–b) The first row shows the drag coefficient of the inversion run including a linear Budd friction law with $m = 1$ and $N_{op}$ (a), as well as $N_{CUAS}$ (b). Both evaluated at the $\lambda_{best} = 1$ of the L-curve using $m = 1$ and $N_{CUAS}$ for better comparison. (c–d) The second row describes the drag coefficient of the inversion run using a non-linear Budd friction law with $m = 3$ and $N_{op}$ (c), as well as $N_{CUAS}$ (d). Again both are evaluated at the $\lambda_{best} = 0.562$ value of the $m = 3$ and $N_{CUAS}$ experiment. Note the different units of the first and second rows.

look at Thwaites Glacier a more ribbed structure can be identified including $N_{op}$ for both linear $m = 1$ and non-linear sliding $m = 3$. The intention is to use the subglacial hydrology model CUAS-MPI to obtain an effective pressure $N_{CUAS}$ that captures most of the structure of the hydrology at the base, making $k^2$ smoother than with a Weertman friction law or the often used effective pressure from geometry like $N_{op}$. This would allow us to interpret the basal drag coefficient $k^2$ in terms of physical properties of the subsurface rather than basal hydrology, since the hydrology is already included in the model. Overall, the $k^2$

in Fig. 11 for both experiments using $N_{CUAS}$ show a higher order of magnitude than the distribution of $k^2$ using $N_{op}$. This can





be explained by the significantly higher magnitude of $N_{\mathrm{op}}$ over the whole study domain compared to $N_{\mathrm{CUAS}}$ (compare Fig. 5). This difference in magnitude is likely reflected in a higher basal drag coefficient $k^2$ for the $N_{\mathrm{CUAS}}$ experiments.

We can evaluate the variance $\sigma^2(\ln(k^2))$ (Table 1, first column) to analyze a scale-independent measure of the structure generated by the basal drag inversion as described in Wolovick et al. (2023). When including effective pressure in terms of
$N_{\mathrm{op}}$ into the linear friction law a decrease of variance can be recognized in comparison to Weertman. But when looking at non-linear sliding an increase in variance happens. In contrast, incorporating $N_{\mathrm{CUAS}}$ a decrease of variance for both linear and non-linear sliding occurs.

Compared to Wolovick et al. (2023), we get relatively high values for the computation of the velocity equivalent of the $J_{\mathrm{obs}}$ misfit (Table 1, seventh column). This could be justified by the higher-order equations which are used in this study or even due
to our higher $\lambda_{\mathrm{best}}$ values. By incorporating effective pressure into the linear friction law, we can report an increase in velocity misfit of $\sim 1\,\mathrm{m\,yr}^{-1}$ for $N_{\mathrm{CUAS}}$ and $\sim 4\,\mathrm{m\,yr}^{-1}$ for $N_{\mathrm{op}}$. In the non-linear case, we instead have a decrease in velocity misfit for the Budd-friction laws compared to Weertman sliding. Especially when including $N_{\mathrm{op}}$ the velocity misfit is $\sim 10\,\mathrm{m\,yr}^{-1}$ less compared to Weertman. An investigation of the total variance of the basal drag inversion shows us the overall quality of the different experiments for the grounded domain (Table 1, column eight). Here, we observe a decrease when including effective
pressure into the friction law for both cases, linear and non-linear sliding.

### 3.3 Non-linear versus linear sliding

In the following section, the effect of a linear vs. a non-linear friction law on the L-curves is examined and emphasized with statistical values from Table 1. Figure 12 shows the L-curve for the linear Budd friction law $m = 1$, $N_{\mathrm{CUAS}}$ experiment (blue color) parallel to the non-linear Budd-type friction law $m = 3$, $N_{\mathrm{CUAS}}$ experiment (red color). To get an impression of the
different $\lambda$ ranges used for linear and non-linear sliding, we also show the L-curve of the $m = 3$, $N_{\mathrm{CUAS}}$ experiment in Fig. 12 with the shifted $\lambda$ range $\left[10^{-2}, 10^{4}\right]$ (gray color), which is also applied for the $m = 1$, $N_{\mathrm{CUAS}}$ experiment. The results show steeper L-curves for the case of non-linear sliding than for the L-curves results of the linear friction laws (Fig. 12). It can be recognized, when comparing Fig. 9b–d to Fig. 10b–d that this could be explained by the increasing $J_{\mathrm{obs}}$ term and the more attenuated $J_{\mathrm{reg}}$ cost function term. This also fits with the finding in Fig. 6 that all L-curves in the second row ($m = 3$) point
towards a higher regularization cost $J_{\mathrm{reg}}$. This was also found in the publication of Wolovick et al. (2023). If we consider Fig. 6, the corner region is characterized by the $\lambda_{\max}$ to the $\lambda_{\min}$ value, which divides the L-curve in the flat $\lambda$-limb and the vertical $\lambda$-limb. We can also conclude that the L-curves in the first row for $m = 1$ have a wider range of their corner regions, which not only contain lower $\lambda$ values but also higher ones, compared to those from the second row for $m = 3$. With a smaller range from $\lambda_{\min}$ to $\lambda_{\max}$, it is easier to select an optimal $\lambda$ value, which also increases the reliability of the L-curves. This scenario
is given by the experiments with a larger non-linearity of $m = 3$ exhibiting a steeper behavior. In addition, the convergence in Fig. 7 displays that the total cost function $J$ gets smaller at least for using Weertman or $N_{\mathrm{CUAS}}$ than the cost function for the non-linear friction law.

When comparing the behavior of linear sliding $m = 1$ with non-linear sliding $m = 3$, we recognize for Weertman, as well as for Budd-type a decrease of variance in $\ln(k^2)$ (Table 1). The decrease is especially high when considering Weertman and Budd



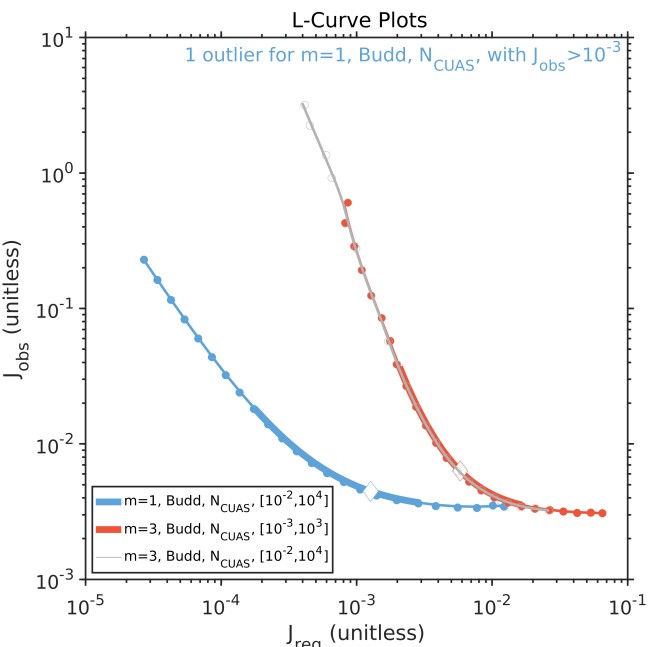

**Figure 12.** L-curve comparison plot for linear sliding $m = 1$ and non-linear sliding $m = 3$ with $N_{\text{CUAS}}$. The blue color illustrate the L-curve for linear sliding and $N_{\text{CUAS}}$ with $\lambda \in [10^{-2}, 10^{4}]$. The red color shows the L-curve for the non-linear friction law with $N_{\text{CUAS}}$ and $\lambda \in [10^{-3}, 10^{3}]$. For a better comparison the L-curve for non-linear friction law with $N_{\text{CUAS}}$ in a range $[10^{-3}, 10^{3}]$ is also given in gray. The dots explain the different inversion runs, the solid line indicates the smooth trade-off curve, the thicker line describes the corner region, and the white, filled diamonds mark the $\lambda_{\text{best}}$ for all shown L-curves. One inversion run for the L-curve with linear friction law ($m = 1$) including $N_{\text{CUAS}}$ detected as outlier.

sliding including $N_{\text{CUAS}}$. For $m = 1$ using $N_{\text{CUAS}}$ we get velocity errors of $\sim 23\,\text{m}\,\text{yr}^{-1}$, whereas we can determine an increase to $\sim 28\,\text{m}\,\text{yr}^{-1}$ when considering a non-linear friction law. But overall, we recognize the strongest increase in velocity misfit of $\sim 10\,\text{m}\,\text{yr}^{-1}$ when using a non-linear Weertman friction law instead of a linear one. In total, this demonstrates that only the velocity misfit of the friction law including $N_{\text{op}}$ decreases when changing from linear to non-linear sliding. Considering the variance ratio of $\ln(k^2)$ and $\boldsymbol{\tau}_{\text{b}}$ we get an overall decrease when using a non-linear friction law. Figure 11 shows more
of the ribbed features in the first row using a linear friction law than for the non-linear friction law in the second row. When comparing the total variance ratio of our basal drag inversion (Table 1, seventh column), we get an improved performance for non-linear sliding independent of Weertman or Budd-type reflected by a strong decrease of total variance ratio.

### 3.4  Best drag estimate

In this section, we focus on the non-linear Budd-type friction law with $N_{\text{CUAS}}$, and $\lambda_{\text{best}} = 0.5$ as it represents the best estimate
of basal drag $\boldsymbol{\tau}_{\text{b}}$ and basal drag coefficient $k^2$ that we achieved (Fig. 6f). We analyze the spatial variability of these fields and conduct comparisons of $\lambda_{\text{min}}$, $\lambda_{\text{max}}$ and $\lambda_{\text{best}}$ from the associated L-curve (Fig. 6f) for the basal drag coefficient $k^2$, the basal



(d) Best basal drag map, m=3, Budd, N$_{CUAS}$

(a) Zoom-in Pine Island Glacier    (b) Zoom-in Thwaites Glacier    (c) Zoom-in Siple Coast

**Figure 13.** Map of best estimated basal drag result $\tau_b$ from WAIS inversion using $\lambda_{best} = 0.5$ and non-linear Budd sliding $m = 3$ incorporating $N_{CUAS}$. The distribution is given in the unit kPa within a range of $[0, 600]$. (a–c) display the zoom-in boxes for Pine Island Glacier, Thwaites Glacier and the Siple Coast as shown in Fig. 1.

drag $\tau_b$ and the velocity error $v_s - v_s^{obs}$ given by a symmetric logarithmic scale (Fig. 15). It should be noted that we do not use the $\lambda_{best} = 0.5$ to show the fields in Fig. 15 but instead the nearest neighbor inversion of $\lambda_{best}$, as well as of $\lambda_{min}$ and $\lambda_{max}$ from the resulting L-curve in Fig. 6f.

Figure 13d shows the resulting spatial distribution of basal drag $\tau_b$. As expected, all glaciers with fast-flowing areas (Fig. 2e) are characterized by a relatively low basal drag (Fig. 13a–c). This can be recognized in the Pine Island Glacier, as well as at the Siple Coast (Mercer, Whillans, Bindschadler and MacAyeal Ice Streams, compare Fig. 1). The Pine Island Glacier and



(d) Best basal drag coefficient map, m=3, Budd, $N_{CUAS}$

**Figure 14.** Map of best estimated basal drag coefficient result $k^2$ from WAIS inversion using $\lambda_{\text{best}} = 0.5$ and non-linear Budd sliding $m = 3$ incorporating $N_{\text{CUAS}}$. The distribution is given within a range of $[0, 6]\,(\text{m}\,\text{yr}^{-1})^{-1/3}$. (a–c) display the zoom-in boxes for Pine Island Glacier, Thwaites Glacier and the Siple Coast as shown in Fig. 1.

MacAyeal Ice Stream display a rippled structure (Fig. 13c). Thwaites Glacier is particularly prominent in this regard, as the structure of the basal drag alternates between high and low basal drag (Fig. 13a). This particular structure is clearly recognizable in the basal drag coefficient in Fig. 14a and could be caused by the underlying bed topography, which reveals a bumpy terrain (Fig. 2a). Overall, Fig. 14d shows a low drag coefficient over the entire study area, apart from the Kamb Ice Stream and Roosevelt Island, and areas further upstream where slow-flowing areas dominate. The Kamb Ice Stream exhibits a very high





**Figure 15.** Map of best drag coefficient and best drag estimate, as well as the velocity error from left to right for the minimum acceptable $\lambda_{\min}$, $\lambda_{\text{best}}$ and maximum acceptable $\lambda_{\max}$ of the experiment with non-linear Budd-type sliding including $N_{\text{CUAS}}$. Plots (a–c) show the basal drag coefficient $\log 10(k^2)$ in $(\text{m yr}^{-1})^{-1/3}$ within a range of $[-4, -0.5]$. (d–f) display plots for basal drag $\boldsymbol{\tau}_b$ in kPa within a range of $[0, 600]$. Figures (g–i) illustrate the velocity error on a symmetric logarithmic scale in $\text{m yr}^{-1}$.

drag, which is in line with the finding of Joughin and Tulaczyk (2002) and Beem et al. (2014) that the glacier is frozen to the bed.

When comparing our results for $\lambda_{\min} = 0.0562$, $\lambda_{\text{best}} = 0.562$ and $\lambda_{\max} = 10$ in the first row of Fig. 15, we can observe in the results the effect of the different weighting of regularization. The basal drag coefficient $k^2$ for $\lambda_{\max}$ displays a really smooth distribution with barely any structure. On the other hand, $\lambda_{\min}$ shows a patchy structure with possible artefacts. This occurrence can also be recognized in the same way in the basal drag $\boldsymbol{\tau}_b$ (Fig. 15d–f). In addition, Plot 15i has a rather high velocity error $(\boldsymbol{v}_s - \boldsymbol{v}_s^{\text{obs}})$ widespread on the entire domain. However, this is not surprising, as $\lambda_{\max} = 10$ provides a high weighting for the



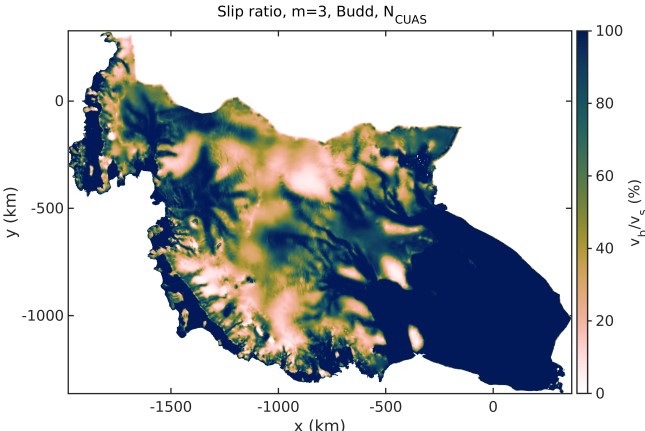

**Figure 16.** Map of slip ratio between basal and surface velocity $v_b/v_s$ for the experiment with non-linear Budd-type sliding including $N_{CUAS}$ at $\lambda_{best} = 0.5$.

regularization term $J_{reg}$, which is minimized together with the first velocity misfit cost term $J_{obs}$. Therefore, the velocity misfit can not get as small as in Fig. 15g. Striking here is the large velocity error at the edge of the whole domain in Fig. 15h,i. Overall, the velocity misfit results in a patchy distribution of both overestimated surface velocities in red color, as well as underestimated modeled surface velocities $v_s$, indicated by blue color.

     Figure 16 displays the ratio of basal and surface velocity, which provides us with insights into the creep-to-sliding behavior

of WAIS. The result shows us that a high proportion of velocity is caused due to sliding at the base (blue colors) exactly in those locations where high velocities (Fig. 1) and a low basal drag $\tau_b$ (Fig. 13d) predominate. When focusing on Thwaites Glacier again a ribbed structure of velocities caused by sliding and creep is visible. Another striking feature is the high creep percentage in some locations of the Thwaites grounding line which matches with the high basal drag (Fig. 13b). Assuming that Roosevelt Island is frozen to the bed, the high proportion of velocity represented by creep (red colors) seems to be very

suitable at this location. The high percentage of creep in the area of Kamb Ice Stream also corresponds with the expectation of Kamb sticking to the bed.

## 4   Discussion

### 4.1   L-curves

We observe different problems in our L-curve procedure, e.g., many outliers for $\lambda < 10^{-2}$ when using linear friction laws and

convergence problems for non-linear sliding. Most of the problems could be solved by shifting the $\lambda$ range towards higher values, further smoothing the initial drag coefficient $k_{init}$ or by choosing different values for the convergence criteria $\Delta_{x_{min}}$ and $\epsilon_{gttol}$ in Eq. (9). As mentioned before, the problem of outliers for too small $\lambda$ values is due to the fact of non-convexity.





It would be desirable to choose in the future one $\lambda_{\mathrm{best}}$ value for the entire Antarctica. But the results with our L-curve procedure show a range of different $\lambda_{\mathrm{best}}$ values, where it gets hard to pick one $\lambda_{\mathrm{best}}$ that fits for the whole domain. One way to
overcome this issue is to apply a method as in Wolovick et al. (2023), where different experiments are used to find an overall best drag distribution. Compared to the values for the $\lambda_{\mathrm{best}}$ values of the Filchner-Ronne region in Wolovick et al. (2023), we obtain even higher $\lambda_{\mathrm{best}}$ values for the WAIS region. But one should notice here, that our experiments are based on the higher-order equations, and their inversion runs are based on the shelfy-stream approximation (SSA equations). Simulating our region with the SSA equations (not shown here) yields $\lambda_{\mathrm{best}}$ values of the same order of magnitude as described in Wolovick
et al. (2023). This fact would imply that an SSA inversion can resolve finer structures than an HO inversion.

The analysis of the influence of linear and non-linear sliding shows us that we get flatter L-curves for $m = 1$, but steeper L-curves for $m = 3$ and with that a smaller corner region, which makes it easier to choose an optimal $\lambda$. We also carried out an L-curve analysis with a linear friction law for a range of $\lambda \in [10^{-3}, 10^{3}]$, which results in an even flatter L-curve.

### 4.2   Utilizing hydrology models in inverse modeling

Here, we discuss the influence of effective pressure and how our results differ for linear and non-linear Budd and Weertman sliding. Like Wolovick et al. (2023), we can argue that our results (Sect. 3.2) suggest that it is reasonable to use a non-linear friction law for a basal drag inversion. Although our results show a strongly increased velocity misfit $J_{\mathrm{obs}}$ compared to Wolovick et al. (2023), which even increases with non-linear sliding, except for using $N_{\mathrm{obs}}$, we see an overall improvement when $m = 3$ is used. We observe a reduction by half of the total variance (Table 1, seventh column) when switching from linear $m = 1$ to non-
linear sliding $m = 3$. This gives us a measure of the overall performance of our inversion, whereby the increased velocity costs in $J_{\mathrm{obs}}$ are compensated by the reduction in the variance of $\ln(k^2)$. Beyond that, when using non-linear sliding, our L-curves indicate a steeper shape than those arising from linear friction laws (Fig. 6, Fig. 12), something also observed by Wolovick et al. (2023). This stronger curve leads to a smaller acceptable corner range for $\lambda$ (Fig. 6) and thus to an easier determination of an optimal $\lambda_{\mathrm{best}}$. In addition, our experiments show that for non-linear sliding using Weertman or Budd including $N_{\mathrm{CUAS}}$ the
convergence of the optimization process is achieved in significantly fewer iteration steps (Fig. 7).

We agree with Wolovick et al. (2023) regarding the non-linearity of friction laws used in basal drag inversion, but our results concerning the use of effective pressure $N$ differ somewhat. The location of the L-curve for the linear and non-linear Weertman case shows a slightly better performance than the L-curves for Budd sliding with $N_{\mathrm{CUAS}}$ (Fig. 10, Fig. 9). However, the velocity misfit for non-linear sliding is reduced from $30.70 \mathrm{myr}^{-1}$ to $27.98 \mathrm{myr}^{-1}$ when switching from Weertman to Budd including
$N_{\mathrm{CUAS}}$. On top of this result, we get the best performance using a non-linear Budd friction law incorporating $N_{\mathrm{CUAS}}$ when looking at the total variance ratio (Table 1). McArthur et al. (2023) obtain a positive correlation between the effective pressure $N$ and the basal drag coefficient $k$ when using a Schoof sliding law, but not when they refer to a Budd-type sliding law. This differs from our results, as we obtain a positive correlation also when using the Budd-type sliding law (compare Table 1). Regions with a lower effective pressure $N_{\mathrm{CUAS}}$ also exhibit a lower basal friction coefficient $k$, which again demonstrates the
controlling role of the hydrological system (compare Fig. 5b and Fig. 14). The slightly increased correlation of $N_{\mathrm{CUAS}}$ to $k_{\mathrm{W}}^2$ compared to $N_{\mathrm{op}}$ to $k_{\mathrm{W}}^2$ shows us that $N_{\mathrm{CUAS}}$ does not need to produce too much structure when computing $J_{\mathrm{obs}}$. This is further



supported by our result of Fig. 11, which demonstrates that $N_{\text{CUAS}}$ leads to a smoother basal drag coefficient $k^2$ distribution, which is our goal when including $N_{\text{CUAS}}$ from a subglacial hydrology model. This confirms that the non-linear Budd friction law with $N_{\text{CUAS}}$ performs well with our model setup. Equally, the results of McArthur et al. (2023) show a smoother basal

drag coefficient, when the effective pressure of a subglacial hydrology model is included. Such a smooth and only slightly variable basal drag coefficient is desirable, because the effective pressure field $N$ should account for the entire structure by including the subglacial hydrology (recall $k_{\text{W}}^2 = k_{\text{B}}^2 N$). The confidence in this experiment $m = 3$, $N_{\text{CUAS}}$ is further supported by the experience of better convergence and the resulting smoother L-curves when including $N_{\text{CUAS}}$. Overall, these key findings justify our argumentation in Sect. 3.4 to define the basal drag $\boldsymbol{\tau}_{\text{b}}$ and drag coefficient $k^2$ as our best estimate when using the

experiment for $N_{\text{CUAS}}$, $m = 3$. Overall, our results, as well as the study of McArthur et al. (2023), demonstrate a more realistic ice sheet model representation when using output of subglacial hydrology models, revealing the importance of coupling ice sheet models with subglacial hydrology models.

### 4.3   Comparison with previous studies/findings in the WAIS

In this section, we compare our best drag and best drag coefficient with other possible lake candidates and findings from

previous studies. When analyzing our best estimate of basal drag $\boldsymbol{\tau}_{\text{b}}$ and basal drag coefficient $k^2$, we find high variability in the Thwaites region (Fig. 13b, Fig. 14b) and high velocities are predominant (Fig. 2e). Here, we observe the characteristic rib-like patterns, so-called *traction ribs* (Sergienko and Hindmarsh, 2013), which varies between high basal drag $\boldsymbol{\tau}_{\text{b}}$ around $200\,\text{kPa}$ and very low regions of drag close to $0\,\text{kPa}$. Rib-like features are also observed in paleo-ice streams (Stokes et al., 2016). Stokes et al. (2016) suggest that those ribbed bedforms found under the ice masses could be caused due to a topographic

expression. Comparing our results with existing seismic measurements of the glacier bed reflection in the Thwaites region could provide us with further insights in the future. However, when we examine the Siple Coast for our best maps of basal drag (Fig. 13c) and basal drag coefficient (Fig. 14c) we have low variability, especially for the Mercer Ice Stream, as well as the Van der Veen and Whillans Ice Stream. Nevertheless, when applying SSA instead of the HO equations, it would be possible to find a higher variability in this region due to the lower $\lambda_{\text{best}}$ values we obtain when using SSA. In that case, the structure might not

be smoothed as much as we observe here at higher $\lambda_{\text{best}}$ values. The Pine Island Glacier in Fig. 13a and Fig. 14a is represented by a relatively low basal drag and basal drag coefficient compared to the drag obtained from Morlighem et al. (2010). While our drag is quite equally distributed, the basal drag for the Pine Island Glacier in Morlighem et al. (2010) shows a very patchy structure, but this could also caused by the higher mesh resolution used in the study.

    Rathmann and Lilien (2022) assume that sticky spots are difficult to detect and bed bumps could be misinterpreted when

deriving the basal drag coefficient using Glen's flow law (Eq. 2). Indeed, some sticky spots identified in the literature, e.g., the sticky spot of Kamb Ice Stream (Luthra et al., 2017), are not visible in our resulting basal drag coefficient field (Fig.14c), nor in the basal drag field (Fig. 13c). We hypothesize, that this could be due to the neglect of crystal-orientation fabric when inferring the basal drag coefficient $k$. However, the sticky spot could also be obscured by the regularization process, the lower mesh resolution in this region (Fig. 3b), or, in general, the fact that we consider a Budd-type sliding law incorporating an

effective pressure field instead of using only Weertman sliding like Rathmann and Lilien (2022). In the latter case, the sticky





spot could already occur in the effective pressure field, which is not recognizable in our case (compare Fig. 5). Overall, we cannot argue, if our basal drag coefficient would significantly change, when considering anisotropic ice instead of isotropic ice as with Glen's flow law. In addition, the choice of flow law, e.g. using an anistropic flow law or including an enhancement factor instead of using Glen's flow law (Eq. 2), could also further impact our results regarding the deformation to sliding ratio
(Fig. 16). Considering our map in Fig. 16, it becomes apparent that most of the glaciers and ice streams of our study domain are controlled by sliding-dominated flow. Comparing these results with those of McCormack et al. (2022), who show that the experiments with ESTAR and Glen with the enhancement factor $E = 5$ (experiment $G_5$) reveal an increase in deformation-induced flow especially in the defined deformation sliding zone and the bed-parallel shear deformation zone, our results would possibly change if the ESTAR flow law or at least an enhancement factor is included in Glen's flow law.

According to Kyrke-Smith et al. (2018), our inferred basal drag conditions should be interpreted with caution, as they recommend separating between skin and form drag. When performing the inversion with detailed bed topography data, they observed a reduced skin drag. Indicating, that the results could include drag, which is due to unresolved topography rather than inherent bed and sediment conditions. Uncertainties in the basal topography could contribute to errors in the inferred basal drag. In addition, Kyrke-Smith et al. (2018) suggest to base the inversion in areas, where traction ribs can be found, like in
the Thwaites region, on high-resolution topography data, not done here. However, Schroeder et al. (2013) investigate the water system transition beneath Thwaites glacier based on a geophysical analysis. They found concentrated channels (no effect on basal drag) near the grounding line, followed by a transition to distributed channels (reducing basal drag) further upstream. Comparing our basal drag distribution with these findings agrees well (Fig. 13b), as we observe a distribution of low drag further upstream that transitions into a distribution of higher drag near the grounding line.

The inversion method and L-curve analysis presented could, of course, be extended to other iterative schemes, as demonstrated in Zhao et al. (2018), or to flow-rate factor inversions (e.g., Arthern et al., 2015; Ranganathan et al., 2021; McArthur et al., 2023). Various studies suggest performing a combined inversion of basal drag and flow-rate factor, e.g., Rathmann and Lilien (2022) recommends such joint inversion to reduce mass-flux errors by compensating for missing fabric information when using Glen's flow law. The simultaneous determination of the flow-rate parameter and the basal drag coefficient would
be necessary, as the ice velocity is controlled by both. When performing a single inversion, like our basal drag inversion, we have to assume an erroneous rheology, and thus, we insert uncertainties into the resulting basal conditions (Ranganathan et al., 2021). However, we argue that both, an iterative scheme and joint inversion, need a good temperature-depth profile to get a good initial state after inversion, as e.g., Zhao et al. (2018) aims for, or to get a good joint inversion result. This is the reason, why we restrict our inversion to an absolute misfit term in the cost function $J$. As a inclusion of the logarithmic misfit term
(e.g., Morlighem et al., 2013) would perform better, if a rheology inversion is considered. The observed surface velocities in the interior of the study area are relatively slow and without proper choice of the rheology (temperature) the velocity contribution from ice deformation could already lead to a higher surface velocities than observed. In this case, the basal drag inversion could not correct for this bias and the logarithmic misfit in these regions would likely be very high. But overall, changing the misfit term does not change the presented approach for the inversion of the basal drag with the used L-curve analysis. However,
incorporating more knowledge into the calculation of rheology, such as a better temperature-depth distribution, could help in





future to obtain better results. Nevertheless, our aim was not to achieve the best steady state, as e.g., Zhao et al. (2018). Our statement is, that focusing on a smooth L-curve result in an inversion is also of great importance, as we experienced that any ill-formed L-curve was always due to difficulties in the inversion process, either caused by the input data or the numerics involved. For example, if the $\lambda$ value is not selected in the corner of the "L", whether due to an incorrect representation, e.g.

not based on log-log scale, or due to a subjective choice of the $\lambda$ value, we would produce unrealistic artefacts in the basal conditions.

**Comparison with subglacial lakes**

The hydrology beneath the glacier plays a major role as a contributor to sliding at the base and, with that, to the velocity of the glaciers. Lakes beneath the ice sheet serve as a lubricant at the ice base. The white circles displayed in Fig. 17 and Fig. 18

describe positions of WAIS where lake candidates from various observational measurements (Carter et al., 2007; Blankenship et al., 2009; Wright and Siegert, 2012; Malczyk et al., 2020) are expected. We compare our best estimate of basal drag $\tau_\mathrm{b}$ for non-linear Budd sliding including $N_\mathrm{CUAS}$ (compare Sect. 3.4) and the same maps for non-linear Budd sliding but using $N_\mathrm{op}$ with these possible subglacial lakes. Examination of the Thwaites Glacier (Fig. 17b) shows for the basal drag $\tau_\mathrm{b}$ using $N_\mathrm{CUAS}$ a relatively good match of low basal drag and lake candidates except for the outlying lake further downstream. The low values

of drag, including the $N_\mathrm{op}$ pressure (Fig. 18a), also match fairly well with the possible lakes. It is clearly visible that even the lake shapes correspond very well with the ribbed structure that can be recognized in the area of the Thwaites Glacier for basal drag incorporating $N_\mathrm{op}$ except for the lake further downstream. The isolated lake further upstream of Thwaites marked with a white circle agrees again with a low basal drag distributionincluding $N_\mathrm{op}$.

When analyzing the Siple Coast in total (Fig. 17c–d, Fig. 18c–d), most of the possible lake candidates that lay in the fast-

flowing regions match quite good with areas of low drag values for both basal drag results using $N_\mathrm{op}$ or $N_\mathrm{CUAS}$. Overall, when comparing Fig. 17a and Fig. 18a, it is striking that the basal drag $\tau_\mathrm{b}$ have a similar distribution of low and high drag values regardless of whether $N_\mathrm{CUAS}$ or $N_\mathrm{op}$ are included, as already assumed in Sect. 3.2. This leads to a relatively good agreement of low basal drag values with the lake candidates in both maps. However, if we compare $k^2$ using $N_\mathrm{CUAS}$ and $k^2$ including $N_\mathrm{op}$ (not shown in the manuscript), it is of course noticeable that the lake candidates match the low values of the basal drag

coefficient with $N_\mathrm{op}$ much better than those of basal drag coefficient with $N_\mathrm{CUAS}$. The different magnitudes of $k^2$ with $N_\mathrm{CUAS}$ and $k^2$ with $N_\mathrm{op}$ are probably due to the different magnitudes of the two effective pressure maps (compare Fig. 5), as already described in Sect. 3.2 and discussed in Fig. 11. Therefore, it is more efficient to just compare the lake candidates with basal drag, when incorporating a Budd-type friction law. It is also noticeable that both basal drag, as well as basal drag coefficient have a different structure when using $N_\mathrm{op}$ or $N_\mathrm{CUAS}$. In $\tau_\mathrm{b}$ including $N_\mathrm{op}$, clearly finer structures can be recognized, both in

the Thwaites region, as well as at the Siple Coast (Fig. 18b–d). Here, ribbed structures are much more visible, which are more smoothed out in the basal drag with $N_\mathrm{CUAS}$ (Fig. 17). As mentioned in Sect. 3.2, this could be explained by the different $\lambda_\mathrm{best}$ values. Because with a higher $\lambda_\mathrm{best}$, as this is the case for the maps including $N_\mathrm{CUAS}$ with $\lambda_\mathrm{best} = 0.5$, the basal structure is smoothed significantly more than with a lower weight as for the maps with $N_\mathrm{op}$ using a lower $\lambda_\mathrm{best} = 0.1$. What should also be







**Figure 17.** Best estimate basal drag map evaluated at its $\lambda_{\text{best}} = 0.5$ value with subglacial lake candidates. (a) shows the basal drag map from Fig. 13d with the possible lakes. (b–d) display zoom-in boxes of basal drag for Thwaites Glacier, Mercer and Whillans Ice Streams, as well as for the MacAyeal Ice Stream. The lake candidates are marked with white lines and circles. These data are used from the datasets of Carter et al. (2007), Blankenship et al. (2009), Wright and Siegert (2012) and Malczyk et al. (2020).

noted is that the structures that both $\tau_{\text{b}}$ using $N_{\text{op}}$ exhibit, correspond very well with the lake candidates and in the Thwaites

region even with the shapes of the three individual lakes further upstream.

Overall, we can achieve a good correlation of the possible lake candidates with our shown maps of $\tau_{\text{b}}$ (Fig. 17, Fig. 18). This agreement for $N_{\text{op}}$ with $\lambda_{\text{best}} = 0.1$ is even better than with $N_{\text{CUAS}}$ with $\lambda_{\text{best}} = 0.5$, which is due to the different degree of smoothing by the regularization. If we consider $N_{\text{CUAS}}$ with the $\lambda_{\text{best}} = 0.1$ value of $N_{\text{op}}$ (not shown in the manuscript), we get similarly good results with more visual features matching the lake candidates and their shape.



**Figure 18.** Basal drag map evaluated at its $\lambda_{\text{best}} = 0.1$ value with subglacial lake candidates. (a) shows the basal drag map $\boldsymbol{\tau}_{\text{b}}$ using non-linear Budd sliding including $N_{\text{op}}$. (b–d) display zoom-in boxes of basal drag for Thwaites Glacier, Mercer and Whillans Ice Streams, as well as for the MacAyeal Ice Stream. The possible lakes, marked with white lines and circles, are used from the datasets of Carter et al. (2007), Blankenship et al. (2009),Wright and Siegert (2012) and Malczyk et al. (2020).

## 5 Conclusions

In this paper, we analyzed in total six basal drag inversion experiments for a large part of WAIS using both, linear and non-linear Weertman, as well as Budd-type friction laws with two different effective pressure descriptions. We especially focused on a basal drag inversion using a Budd-type friction law incorporating an effective pressure from a hydrology model to improve the basal drag field for a major part of WAIS. We conclude, how to deal with poorly shaped L-curves for weighting a regularization term and ended up with six very smooth ones. The L-curve analysis taught us that it is essential to not accept an ill-shaped L-curve with many outliers, but to find the inconsistencies in the model setup. In all cases, we were able to achieve a smoother



L-shaped curve, if different actions, e.g., shifting the $\lambda$-range, have been made. The subdomain L-curve analysis reveals that subdomains with different geometry settings have an effect on the shape and smoothness of the L-curves. Different problematic areas can also be recognized by means of many outliers.

Our results suggest, as previous studies, that it might be useful to rely on a non-linear friction law when using basal drag inversions. We demonstrate that including effective pressure from the subglacial hydrology model CUAS-MPI can further improve the resulting maps of a basal drag inversion. We were able to show that the incorporated effective pressure field explains a significant fraction of the variance in drag coefficient. Even though we believe that we could achieve even better results for the basal drag coefficient by further improving the effective pressure of CUAS-MPI. Our resulting best estimate of

basal drag and basal drag coefficient distribution reveals a good fit with data of possible lake candidates. The ribbed structure that we recognize in parts of Thwaites in the drag coefficient maps could be confirmed in the future with seismic measurements. As we have high confidence in our results, our achieved basal drag can serve as an initial stress state for further models considering a major part of WAIS. In the future, the behavior of the L-curves and their analysis should also be compared with the other areas of Antarctica or even the entire Antarctic.

*Code and data availability.*   The used open-source Ice-Sheet and Sea-level System Model (ISSM v. 4.22; Larour et al. (2012)) is available at https://issm.jpl.nasa.gov/. The inversion scripts are available at https://doi.org/10.5281/zenodo.7798650 (Wolovick et al., 2023). All parameters that were adjusted for this study, as well as all used scripts to create the figures are submitted at https://doi.org/10.5281/zenodo.10974434. The results of the inverse model are saved in NetCDF format and also submitted at https://doi.org/10.5281/zenodo.10974434. For the geometry of our model, as well as for the CUAS-MPI model, we use the BedMachine Antarctica v2 dataset (Morlighem et al., 2020) provided

on https://doi.org/10.5067/E1QL9HFQ7A8M (Morlighem, 2020). The observed surface velocities used from the MEaSUREs v2 dataset (Mouginot et al., 2012; Rignot et al., 2011b) are accessible through https://doi.org/10.5067/D7GK8F5J8M8R (Rignot et al., 2017). For the 1D thermal model we use surface climate inputs of surface temperature (Comiso, 2000) and accumulation rate (mean of Van De Berg et al. (2005) and Arthern et al. (2006)). All those data can be found at https://doi.org/10.1594/PANGAEA.734145, 2010 (Le Brocq et al., 2010; Le Brocq et al., 2010). The output of the 1D thermal model, as well as the effective pressure from CUAS-MPI are provided within a NetCDF

file at https://doi.org/10.5281/zenodo.10974434.

*Author contributions.*   LSH performed the inversion runs with ISSM, prepared the figures and the manuscript text. LSH and MW created the code scripts for the inversion and the figures. TK conducted CUAS-MPI simulations to provide effective pressure. AH detected the shear margins for the mesh. MR installed and maintained the ISSM installation on AWI's HPC system. AH, AR and LSH designed the study. AH and AR supervised the work. All authors contributed to discussions and ideas regarding the manuscript.

*Competing interests.*   The authors declare that they have no conflict of interest.



*Acknowledgements.* The first author Lea-Sophie Höyns is funded through the Helmholtz School for Marine Data Science (MarDATA), Grant No. HIDSS-0005.



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
