# Peer review of "Improved basal drag of the West Antarctic Ice Sheet from L-curve analysis of inverse models utilizing subglacial hydrology simulations"

_EGUsphere, 2024_

## Referee Comment (RC1)

Review for "Improved basal drag of the West Antarctic Ice Sheet from L-curve analysis of inverse models utilizing subglacial hydrology simulations" by Höyns et al.

In this manuscript, the authors analyse ensembles of basal drag inversions with a range of different values for their regularisation parameter, using six different variants of sliding laws; Weertman and Budd sliding with m=1 and m=3, with two different formulations of effective pressure within the Budd sliding law, $N_{op}$ and $N_{CUAS}$. For each sliding law, an L-curve is produced and the authors identify a "best" value for their regularisation, along with maximum and minimum acceptable values. They explore various methods by which to improve the appearance of their L-curves, and carry out a sub-domain L-curve analysis which reveals differences in optimal regularisation values for areas of the domain with different glaciological settings.

Comparisons are made between the outputs produced by the "best" regularisation values in each case, and also between the maximum and minimum values in the authors' preferred case of non-linear Budd sliding using $N_{CUAS}$, for which differences in the structure of the output are discussed. Finally the authors attempt to validate their preferred basal friction field by drawing comparisons with previous studies of bed structures and potential subglacial lake positions.

General comments:

There are a lot of ideas presented in this manuscript, some of which are more convincing to me than others. The model setup and majority of the methodology is thoroughly explained, and entirely appropriate to the topics being addressed.

The main highlight of this manuscript for me was the idea of sub-domain L-curve analysis. The idea that different regularisation may be needed in areas which contain fundamentally different physical settings, and therefore major differences in the ice flow, is an important one. I would have been interested in seeing far more focus on this aspect of the work. Comparisons in those sub-domains of the outputs using the locally-defined best regularisation against using the globally-defined value would have been illuminating, and I think this very promising avenue should be explored more in the future.

Another highlight was the results displayed in Fig.15, showing very clearly why it is important to consider regularisation carefully, and the difference it can make in a model.

In general, I felt there was too much focus on the fine details of L-curve shapes without much explanation of why readers should care about this. The details of the L-curves and

convergence are likely to be specific to ISSM, while other models have different inversion processes, and different regularisation parameters. Some models require choices for multiple parameters, which require multi-dimensional "L-surfaces", and would require entirely different analysis. To my mind, the more interesting and useful comparisons for a wider audience outside of ISSM modellers were made when it was shown what the effect of different regularisation values was on the actual outputs of the inversion (ie. basal traction fields). I wasn't convinced of the importance of how smooth the curves were from which the values came, or the statements being made regarding the comparative shapes of the curves.

I was also not particularly convinced by the comparison with proposed subglacial lake positions. Partly because it is not actually known whether lakes really do exist in these locations, but mostly that I do not see much meaningful correlation between the basal drag fields and the lake outlines in the figures.

This manuscript appears to rely a lot on reference to Wolovick et al. (2023), but I think the most important parts of methodology should be restated here. In particular, the procedure for picking $\lambda_{best}$, $\lambda_{min}$ and $\lambda_{max}$, which are crucial to the results of this manuscript, is buried in an appendix in Wolovick et al. and should be stated clearly to help readers understand the L-curve figures.

Overall, I believe that this manuscript contains some very interesting and important work, but that it is somewhat hidden amongst a wide variety of ideas which could be organised in a better way, and some of which seem specific to the model being used. I think this manuscript could benefit from being edited down to a more concise form, and that some restructuring could benefit its readability.

I would recommend major revisions before this manuscript is accepted for publication.

Specific comments:

Abstract

Lines 10-12: It's a bit unclear what "best" and "worst" mean in terms of L-curve behaviour. Perhaps saying Pine Island produces the smoothest curves would be a better way of wording this?

Lines 16-17: I don't agree with using "more accurate" here. The choice of parameter, as the authors point out, is a balance between matching and regularising the observations. Accuracy could be interpreted as getting the closest fit to observations, which isn't the point being made. The regularisation parameter itself cannot be defined as being accurate or not, as there is no physical real-world comparison to make.

Line 18: "improved performance" in terms of what? Convergence, smoothness of L-curves? This should be specified.

Line 20: Should this be a comma rather than two separate sentences?

Section 1. Introduction

Lines 35-36: Could you expand on what exactly is meant by "the majority of high velocities are cause by sliding", or provide a reference? Is this just saying that sliding causes higher velocities than flow driven by surface gradients, or is it claiming that changes to basal sliding have a larger influence than other processes such as mass balance?

Line 43: I'd say "represent" rather than "compute". The parameters from inversions broadly represent a combination of several factors (as is pointed out in the next paragraph), rather than being a value which physically describes one thing.

Lines 87-89: This reads as if the Budd law doesn't account effective pressure. Should be reworded for clarity.

Lines 98-100: Isn't the inverse problem in glaciological models always ill-posed, regardless of sliding laws or domains? Could this sentence be explain further?

Section 2. Method

Some restructuring in this section could be beneficial, as there appears to be some repetition, and jumping between topics.

Lines 108-115: I don't think this section is necessary. ISSM can be introduced at the start of 2.1 instead, and other parts stated in the relevant sections.

Lines 134-139: This could be moved into 2.3, with the rest of the inversion description.

Lines 166-176: The discussion of B fields seems to interrupt the description of the forward model. It could be better placed in the subglacial hydrology section as it is being discussed in relation to CUAS.

Lines 189-194: Why not introduce both effective pressure parameterisations in the same section (ie. move this to the next section).

Lines 233-236: These two explanatory sentences belong in the introduction (in fact, I think they restate something already in the introduction)

Line 243: Given how important the values of $\lambda_{best}$, $\lambda_{min}$ and $\lambda_{max}$ are for this work, I think the method should be shown here rather than just a reference.

Lines 276-280: Should the detail about the forward model solver be under the Forward Model section?

Section 3. Results

Lines 304-305: Why not just use a full range of $[10^{-3} \, 10^4]$ for all L-curves?

Line 313: How are outliers identified? I assume these are points which lie over a certain distance from the smoothed tradeoff curve, in which case this should be specified. Or are they simply the inversions which struggled to reach convergence? In particular, for Fig. 8(d,g,h) it isn't clear why some of the outliers shown shouldn't be part of the curve.

Lines 318-329: Was the same smoothing of $k_{init}$ and the same values of $\varepsilon_{gttol}$ and $\Delta_{xmin}$ applied for all inversions for all sliding laws? Given that these can have an important impact of the results I assume a like-for-like comparison has been conducted, but it could read as if different smoothing was used in different cases.

Lines 335-342: As a more general point, is "best" the right word to use? The choice, as is always the case with L-curves, is quite subjective. I agree that the identified corner regions are a good guide for picking your regularisation parameter, and appreciate the more rigorous methodology behind picking a value rather than doing so by eye, but I would assume any pick within this region would be reasonable. Specifically, as noted here, in some cases the identified $\lambda_{best}$ is very close to $\lambda_{min}$, and perhaps another argument could be made to say that a value halfway along the curve between $\lambda_{max}$ and $\lambda_{max}$ could be the best choice. Do you have any insight into what variability the choice within the corner region causes in the output? It would be interesting to see the differences between your $\lambda_{best}$ and a value one might pick by eye, to give more of an idea of how much this matters.

Lines 343-345: Is it necessarily to be expected that $\lambda$ values would be close to each other? What were the differences in the other inversions mentioned here which display larger variability, and why should that make them less trustworthy?

Lines 346-354 & Fig.7: The convergence could be specific to the particular inversion process of ISSM with the chosen optimisation algorithm of M1QN3. Appendix B1 of Barnes et al. (2021) shows improved performance (greater minimisation of the cost function) achieved using an interior point algorithm (Byrd et al., 1999). Do you have any thoughts on whether you might find similar differences in convergence between your cases using a different algorithm such as this?

Line 355: I think this section should be 3.2. It contains enough to stand alone. I like the idea of running inversions on subdomains to find whether different regularisation could be required under different physical conditions. I think the values of $\lambda_{best}$ should be highlighted on Fig.8 as they are in Fig.6, to show the difference more clearly. I would personally also be very interested to see what difference would be seen in the inversion outputs going into a forward run, comparing a spatially-varying $\lambda_{best}$ to the global value.

Fig 8: This figure, and some that follow, could be made clearer by highlighting the corner region in a different colour, rather than the thicker line currently used. I assume the circles with white in the middle are outliers, but this is not labelled on the figure. Some of the L-curves presented in this figure could benefit from an extended range of values, as the corners appear very close to the end of the intervals used.

Figs 9,10,12: The diamonds showing location of $\lambda_{best}$ is not clear. The values of $\lambda_{best}$ could also be shown.

Lines 464-471: I'm not sure I follow the reasoning for using the same $\lambda_{best}$ in both cases. The whole point in the inversion process is surely to pick an optimal regularisation for each case, in order that the output is most suited to the individual problem. I would think it was more instructive to compare the outputs that come from each experiment's $\lambda_{best}$ value. Also, the numbers given do not correspond to the $\lambda_{best}$ values stated on Fig.6. If these are coming from different L-curves it should be made clearer.

Fig.11: Why not also show the Weertman output?

Lines 519-520: Could you explain how the chosen case is the "best estimate" of those on Fig.6?

Fig.15: This is a great figure for showing why regularisation choices are important, and the difference they can make!

Section 4. Discussion

Line 558: It may be desirable from a practical perspective, but for better results maybe picking a few different regularisation values based on ice speed or other obvious differences in the physical setting would be the way to go. I find this idea very interesting.

Figs 17,18: I don't personally find these to be particularly convincing. While there are a couple of places where the basal drag field lines up with the white lines, there are also places where the outlines cross areas of higher friction. What are the locations of these "possible" lakes based on? Is there a correlation with their positions in the effective pressure fields $N_{op}$ and $N_{CUAS}$? Is similar correlation seen for the Weertman inversion output as the Budd ones?

References

Barnes, J.M., Dias dos Santos, T., Goldberg, D., Gudmundsson, G.H., Morlighem, M. and De Rydt, J., 2021. The transferability of adjoint inversion products between different ice flow models. *The Cryosphere*, *15*(4), pp.1975-2000.

Byrd, R.H., Hribar, M.E. and Nocedal, J., 1999. An interior point algorithm for large-scale nonlinear programming. *SIAM Journal on Optimization*, *9*(4), pp.877-900.

Wolovick, M., Humbert, A., Kleiner, T. and Rückamp, M., 2023. Regularization and L-curves in ice sheet inverse models: a case study in the Filchner–Ronne catchment. *The Cryosphere*, *17*(12), pp.5027-5060.

---

## Referee Comment (RC2)

**REVIEW: Improved basal drag of the West Antarctic Ice Sheet from L-curve analysis of inverse models utilizing subglacial hydrology simulations**

The authors perform an L-curve analysis on a suite of basal drag inversions applied to a model domain that encompasses much of WAIS. L-curve analysis allows them to examine the balance between cost function terms when different degrees of regularization are utilized in the inversion, represented in the manuscript as a range of $\lambda$ values, where $\lambda$ is the weight associated with the regularization term in the inversion cost function. Inversions are analyzed with respect to the trade off between components of the cost function: $J_{obs}$, associated with velocity misfit compared to observations, and $J_{reg}$, associated with regularization. The best performing inversion is chosen based on the value of $\lambda$ which minimizes the cost function without either component of the cost function growing too large. This is done analytically by finding the maximum curvature of the L-curve.

Suites of inversions are performed under the assumption of linear (m=1) and nonlinear sliding (m=3), and using three treatments of effective pressure: the exclusion of effective pressure (Weertman), effective pressure approximated based on ice geometry (Budd with $N_{op}$), and effective pressure calculated from a subglacial hydrology model (Budd with $N_{CUAS}$). Comparisons are drawn across the resulting 6 experimental groups allowing for analysis of the $\lambda$ values which produce the best inversions when different physics are applied in the model parameterization. Additionally, the authors break their large model domain into 6 smaller subdomains that incorporate distinct geographic characteristics and perform the same analysis. This provides practical insights into how the inversion algorithm functions when applied to various portions of the ice sheet, and illustrates some complications of inverting for basal drag across large portions of the ice sheet without specialized treatment to certain areas.

**Overall Statement:**

On balance, there is a lot of very practical and useful information in this paper. The authors provide a great deal of detail with respect to the methods and L-curve analysis, which makes this work a resource for anyone interested in incorporating a more rigorous analysis to the inversion steps in their modeling procedure. This type of practical resource is not so often published in the glacier modeling community, although it perhaps should be to improve the reproducibility of modeling outputs and provide greater access to learning new methods/improving existing ones. Additionally, the main results of this work provide easy to implement improvements to basal drag inversions across a large portion of the Antarctic Ice Sheet.

My main critiques of this work pertain to the organization of the paper. I believe that the readability of the manuscript could be largely improved by reducing the length and simplifying the language, particularly in the Results section where there seems to be quite a bit of repetition

and bouncing back and forth between ideas. That being said, I think the analysis of this work is sound and interesting and merits publication.

**General Comments:**

The Results section of this paper is 16 pages. While I appreciate the attention to detail, it would benefit from some simplification and reorganization. It is currently divided into subsections, but discussions pertaining to other parts of the study are present in each of the subsections. This causes quite a bit of repetition throughout the Results. I would suggest tightening up each of the subsections to highlight the main result and simplify the message. Comparison of the main results is done quite nicely later in the paper. It doesn't need to be addressed here as well. Readability of the Results section could be further increased by either condensing the description of L-curve analysis and focusing on giving clear interpretations of the figures, or by segmenting off some of these details into a supplemental document.

The text on many of the figures is quite small, making the subplots difficult to interpret. I would suggest making some of the smaller figures larger in general; specifically figures 7, 9, 10, 12, and 16. Some of the markers on subplots are also quite difficult to see. Figures 8 and 15 both illustrate the information being presented very nicely.

The Discussion and Conclusions sections are well written and clearly move through the main conclusions of the paper. However, the connection to subglacial lakes seems a bit out of sync to the rest of the paper, and I am not entirely convinced of the argument being made. It seems to me that correlation between $\tau_b$ and observed lake location is due to the use of N in the friction inversion. I think this section could probably be removed or condensed, but right now is somewhat distracting from the main results of this work.

I would be curious to know how these results might differ when using different forms of friction laws. Weertman and Budd are both power laws, but a regularized Coulomb friction law may produce significantly different behaviors, especially when comparing the performance of the inversion across subdomains. I understand that doing the full analysis using additional friction laws is a large amount of work, and I think the results presented here warrant publication without additional modeling. However, if the authors have a sense of how these results might translate to other friction laws, it would be an interesting point of discussion.

**Specific Comments:**

**Abstract**

Line 2-5: This sentence is overly complex. I would suggest pairing it down as it is just context.

Line 10: It reveals that Pine Island Glacier being → Pine Island Glacier is

Line 11: "best" and "worst" feel like odd descriptors here without more context for the interpretation of the L-curves. Consider changing or adding a bit more detail.

Line 12: Remove the comma

Line 19: "a larger part of the spatial basal drag coefficient structure" is somewhat confusing, consider rephrasing.

Line 20: "Allowing the inverted drag…" This is an incomplete sentence, combine with previous or remove.

Line 21: "reflect actual variations in basal properties" implies comparison to observations. If this is what you intended it should be stated more explicitly, but I don't believe that was part of the study.

**Introduction**

Line 30-31: "The ongoing continuous…" This sentence is awkward and reads as an incomplete thought, consider rephrasing.

Lines 35-45: This paragraph jumps from discussing friction to subglacial hydrology to basal drag inversions. It feels a bit jumbled. I suggest reorganizing to discuss basal friction, the difficulty of parameterizing it, and inversions together first. Then introduce subglacial hydrology as an additional complication in determining basal friction.

Line 58: "However, it would be possible…" Suggest removing this sentence. It is not necessary to make your point.

Line 61-70: This paragraph is good, but it should come earlier. Here you are introducing concepts that are referenced with less context in the previous paragraph as though it is the first time they are being brought up.

The first two paragraphs of the introduction are somewhat disorganized and confusing in the way they introduce the main concepts of this work. But the rest of the introduction flows quite well, is easy to follow, and sets up the objectives of the paper nicely.

**Methods**

Line 118: Remove the comma.

Line 156: "model that models" is repetitive; this sentence should be rephrased.

Line 219: "we set it to ice pressure pi and neglect": I assume that "it" is in reference here to effective pressure, but this sentence would be clearer if that was stated explicitly.

Line 237: To improve this instability of the problem, it is beneficial to regularize it → To improve the instability of the inverse problem, it is beneficial to add regularization.

In general, frequent use of pronouns in technical writing adds confusion. When you can be specific in what you are referencing, it is good to do so.

Line 244-246: Description of $\lambda$ range should be simplified. The distinction of different ranges being used is not necessary here and adds confusion because it is not yet stated when or why each range is applied.

Line 246-249: Not sure that this is necessary.

There is a lot of really useful detail provided in this section that is often left out of modeling papers. I appreciate the inclusion of helpful tips that others in this field might find useful. But I do think that some reorganization could improve the readability and flow of ideas.

***Results***

Line 313: How are outliers defined?

Line 315-317: Confusing sentence, rephrase.

Line 319: The information about smoothing and convergence criteria in this paragraph is really useful!

Line 330: Maybe point to two subplot panels here that highlight the distinction being made by a very flat L-curve

Line 335: curve fitting procedure is → curve fitting procedure, is

Line 338: "cost curvature" is a bit confusing. I suggest rephrasing to "curvature of the cost function" at least the first time it is mentioned.

Line 340: In this method, you are picking $\lambda_{\text{best}}$ to be the $\lambda$ corresponding to the absolute maximum curvature of the L-curve, if I am understanding correctly. If so, I would say so rather than describing the small side peak in the curvature figures.

Line 344: Suggest removing reference to results that are not included in this work or published elsewhere.

Line 345: How are you defining good results? Is this in reference to the $\lambda_{\text{best}}$ range being small? If so, can you provide some citations to other work to back up this claim?

Line 374: Fig. 8b-i $\rightarrow$ Fig. 8b

Line 379: Remove comma

Line 392: Fig. 8c $\rightarrow$ Fig. 8d

Line 398: Conspicuous is that the subdomain $\rightarrow$ It is conspicuous that the subdomain

Line 399: This sentence does not follow from the previous one.

Line 401: I suggest separating the discussion of different geographies and discussion of different friction laws. These last few paragraphs could be moved into the section dedicated to linear vs non-linear sliding.

Line 404: I suggest rephrasing this sentence for clarity.

Line 408: Is there really no recognizable impact? I'm not sure that I agree. There are few outliers with linear sliding, but the different regions have quite different shaped L-curves with different $\lambda_{\text{best}}$ values. Please elaborate to make this point more clear.

Table 1: It would be helpful to have the $\lambda_{\text{best}}$ values included here, or in their own table. These values are referenced quite a lot throughout the paper, but they are not listed anywhere.

Line 433: You state that $\lambda_{\text{best}}$ for the $N_{\text{CUAS}}$ run is located between the two other runs, but this is not clear on the figure. Please provide $\lambda_{\text{best}}$ for all three runs to make this point.

Figures 9 and 10: Subplots are very difficult to read. Text is too small and outlier markers are difficult to see. I would also suggest using colors that will contrast in grayscale.

Line 452: Unclear what "accounting for less structure" means.

Figure 11: This is super helpful!

Line 494: parallel to → alongside

Line 497/Figure 12: Why not show this on the figure too?

Line 499-507: The point being made here is a bit unclear.

Figure 15: Great figure!

Section 3.4: In this section, I'm still wondering if $\lambda_{\text{best}}$ is actually the best choice of $\lambda$, especially since $\lambda_{\text{min}}$ has less error from velocity misfit while still incorporating regularization. I think this section would benefit from reiterating the justification of the choice of $\lambda_{\text{best}}$ directly.

*Discussion*

Line 557: "the problem of outliers for too small $\lambda$ values is due to the fact of non-convexity" I'm unclear about what this means.

Line 562-565: Unclear about how HO is being applied- aren't your experiments using SSA inversions? Where does the HO model come in?

Line 618: could also caused → could also be caused

Line 654: As a inclusion → As an inclusion

The discussion is nicely constructed and overall very clear.

*Conclusions*

Great well written summary!

---

## Author Response (AR1)

**Response to Editor and Reviewers**

Dear Felicity McCormack,

thank you for your remarks. We provided a revised version of our manuscript and explicitly include your comments. In the following, we summarize the response to your comments, detailing where edits in the track changes document belong to the structure and readability, as well as the response to both reviewer comments. In the revised version we made additional adjustments to the sentences of our responses to the reviewers. These updates are highlighted in green for clarity. All other comments and our corresponding responses are marked with blue color.

**Answer to Editor**

There are a couple of points in the reviewers' comments that would be good to see addressed in a bit more detail, as follows:

- R1 makes a good point about the balance between the details of the L-curves (i.e., specifics relevant to ISSM) and those of the effect of different regularisation values on the outputs of the inversion (basal traction fields). The latter is also very relevant as we're starting to see more outputs from advances in automatic differentiation, where small differences in basal traction can make a large difference in the evolution of the system over time, and particularly in response to climate change. It would be good to make sure that aspects of the work that have broad applicability for ice sheet modelling studies is addressed in the discussion.

Yes, we agree that the manuscript was a bit imbalanced. To address this, we removed some of the detailed discussion on ISSM specific L-curve aspects and included additional details on the effects of varying regularization values. Accordingly, we have added the following sentences to the discussion:

"Morlighem et al. (2021) highlight that even small variations in basal friction significantly impact glacier dynamics by using automatic differentiation. In addition, Barnes and Gudmundsson (2022) show that simulations with lower basal friction coefficients exhibit greater sensitivity to changes in forcing. Although their study employs a constant basal drag coefficient to explore this effect, accurately characterizing the sliding laws and the basal drag coefficient remains crucial for reliable predictions of future ice sheet evolution. Our Fig. 14 illustrates how varying λ values influence the simulated basal drag fields, demonstrating again the importance of precise regularization values and the application of a subdomain L-curve analysis in the future."

-I also agree with the reviewer that the point about potentially needing different λbest values for different basal friction regimes is a key result – one of the most significant and compelling results – and one that deserves a more detailed discussion. How do you envisage such an analysis could be undertaken? Can you make recommendations about an approach that could be suitable here, particularly given what is known about the ice-bed interface zone (e.g., from geology and subglacial hydrology)?

Yes, we agree that this represents a key result of our manuscript. To emphasize this, we focused more on the subdomain L-curve analysis and included the following sentences in the discussion to outline a potential approach:

"Another approach for selecting optimal λbest values could involve relying on a subdomain L-curve analysis, as demonstrated in Sect. 3.2. A key finding of this manuscript is that different regularization weights may be required for areas with fundamentally distinct physical conditions, leading to significant differences in ice flow. One potential method would be to determine λbest values using an analysis setup similar to the control field approach (compare Fig. 3a, used for mesh generation with varying refinements). This could include classification based on factors such as ice speed, surface and bed slopes, thickness gradients, or other notable physical differences. Overall, our subdomain L-curve analysis reveals that shear margin regions pose the greatest challenge when using non-linear sliding laws. In contrast, slow-flow areas like the WAIS divide or upstream tributaries of Thwaites Glacier show minimal influence on regularization, regardless of linear or non-linear sliding law. Similarly, rock outcrop regions affect the L-curve less than expected. Fast-flow regions such as Thwaites Glacier, Pine Island Glacier, and the Siple Coast exhibit relatively smooth subdomain L-curves for both linear and non-linear sliding."

**Our References**

Barnes, J. M. and Gudmundsson, G. H.: The predictive power of ice sheet models and the regional sensitivity of ice loss to basal sliding parameterisations: a case study of Pine Island and Thwaites glaciers, West Antarctica, The Cryosphere, 16, 4291–4304, https://doi.org/10.5194/tc-16-4291-2022, 2022.

Morlighem, M., Goldberg, D., Dias dos Santos, T., Lee, J., & Sagebaum, M. (2021). Mapping the sensitivity of the Amundsen Sea Embayment to changes in external forcings using automatic differentiation. *Geophysical Research Letters*, 48, e2021GL095440. https://doi.org/10.1029/2021GL095440

**Track changes**

Track changes that belong to structure and readability: (- Both R1 and R2 comment on structure and conciseness for improved readability. It would be good to please point out where edits in the tracked changes document pertain to this request.)

To emphasize the mentioned revisions related to structure and readability, we have highlighted those sections in the track changes using a yellow marker. We made efforts to shorten the manuscript, particularly the results section, but the length reflects as a result a compromise to also address additional changes and comments, as well as enlarged figures.

**Abstract**
We edit the abstract to enhance readability and included an additional sentence addressing regularization for varying physical properties.

**Introduction**
We reworded and shortened the first paragraphs for improved readability and clarity.

Additionally, we restructured the introduction as recommended by Reviewer 2. The introduction now begins with presenting the general problem, followed by an explanation of friction, a description of the friction law, an overview of subglacial hydrology, and then a discussion of the inverse problem with regularization and the L-curves. We then outline current state-of-the-art approaches, our objectives, and the structure of the manuscript.

**Method**

We reorganized this section for improved readability by avoiding the jumps between topics as recommended by Reviewers 1 and 2. We deleted the introduction of this method chapter to shorten the manuscript. Repetitions already covered in the results or introduction chapter were removed, and some details were relocated to the results section to better emphasize the methodology and enhance the reading flow.

**Results**

In Sect. 3.1 on the L-curve analysis, we focused on significant shortening, improving the structure, avoiding repetition, and reorganizing the chapter as a whole. Additionally, the section on inversion convergence has been moved to the appendix for further shortening.
In the next section on subdomain L-curve analysis, we primarily relocated a larger part to Sect. 3.4 to improve the readability and to avoid jumping between different topics. To highlight one of the key results of the manuscript, we added an additional sentence addressing regularization in different domains.
In the sections on effective pressure and non-linearity, we simplified overly complex sentences, removed less significant content - particularly details about the L-curves and repeated information - and restructured certain parts to enhance both readability and overall structure.
We adjusted one part of the sequence in the best drag estimate section to improve its structure and included a sentence regarding the high drag near the grounding line that was missing in the previous version.
Overall, we have moved some parts of the results section to the discussion section to again improve the flow of reading.

**Discussion**

We incorporated the previously mentioned results into the discussion, reorganized some of the content and refined sentence structures in some parts. Additionally, we addressed specific requests by emphasizing the topic on subdomain L-curves, convergence with other algorithms, the use of alternative friction laws, like the Coulomb friction law, and the behavior of basal drag fields under different regularization values. The subsection on subglacial lakes was removed from the discussion.

**Response to Reviewer 1**

**Review for "Improved basal drag of the West Antarctic Ice Sheet from L-curve analysis of inverse models utilizing subglacial hydrology simulations" by Höyns et al.**

The original reviewer comments are in black and our responses are highlighted with blue color.

**Overall answer to RC1:**

Dear Reviewer,

thank you for all of your remarks. Your suggestions will help us to further improve our manuscript! Even though we were encouraged not to create a revised version yet, it was easier to incorporate the changes already and facilitate the response to your comments, which is why we have already prepared a next manuscript version. We have tried to incorporate most of your comments into this revised version.
Both reviewers suggest major revisions before publishing the manuscript. In the future version we will try to reorganize the structure of the manuscript and make it far more efficient to read, especially for the method and results sections. As both reviews suggest to remove or condense the section on subglacial lakes we decided to remove it from the upcoming manuscript version. In addition, we will work on the presentation of our figures and in particular on the figures showing the L-curves and their cost function curvature. We will add a new part to the discussion section in which we want to discuss our results with regard to the regularized Coulomb friction law. Further, we will add a description of our selection method for the λbest, λmin and λmax of our L-curves.

In this manuscript, the authors analyse ensembles of basal drag inversions with a range of different values for their regularisation parameter, using six different variants of sliding laws; Weertman and Budd sliding with m=1 and m=3, with two different formulations of effective pressure within the Budd sliding law, Nop and NCUAS. For each sliding law, an L-curve is produced and the authors identify a "best" value for their regularisation, along with maximum and minimum acceptable values. They explore various methods by which to improve the appearance of their L-curves, and carry out a sub-domain L-curve analysis which reveals differences in optimal regularisation values for areas of the domain with different glaciological settings.

Comparisons are made between the outputs produced by the "best" regularisation values in each case, and also between the maximum and minimum values in the authors' preferred case of non-linear Budd sliding using NCUAS, for which differences in the structure of the output are discussed. Finally the authors attempt to validate their preferred basal friction field by drawing comparisons with previous studies of bed structures and potential subglacial lake positions.

**General comments:**

There are a lot of ideas presented in this manuscript, some of which are more convincing to me than others. The model setup and majority of the methodology is thoroughly explained, and entirely appropriate to the topics being addressed.

Thank you for your remarks!

The main highlight of this manuscript for me was the idea of sub-domain L-curve analysis. The idea that different regularisation may be needed in areas which contain fundamentally different physical settings, and therefore major differences in the ice flow, is an important one. I would have been interested in seeing far more focus on this aspect of the work. Comparisons in those sub-domains of the outputs using the locally-defined best regularisation against using the globally-defined value would have been illuminating, and I think this very promising avenue should be explored more in the future.

Yes, locally defined best values for regularization is a really interesting point and worth to be explored more in the future. We briefly comment on that in the detailed responses below.

Another highlight was the results displayed in Fig.15, showing very clearly why it is important to consider regularisation carefully, and the difference it can make in a model.

Thank you!

In general, I felt there was too much focus on the fine details of L-curve shapes without much explanation of why readers should care about this. The details of the L-curves and convergence are likely to be specific to ISSM, while other models have different inversion processes, and different regularisation parameters. Some models require choices for multiple parameters, which require multi-dimensional "L-surfaces", and would require entirely different analysis. To my mind, the more interesting and useful comparisons for a wider audience outside of ISSM modellers were made when it was shown what the effect of different regularisation values was on the actual outputs of the inversion (ie. basal traction fields). I wasn't convinced of the importance of how smooth the curves were from which the values came, or the statements being made regarding the comparative shapes of the curves.

We agree that our manuscript is relatively detailed and in some cases particularly important for ISSM users, such as the values specified for the convergence criteria. To show that those details are also suitable for other models one would need to prepare comparisons between those ice-sheet models and algorithms like Barnes et al., (2021) did. Nevertheless, the details of the L-curve analysis, whether on the shape or the regularization of our results, also apply to other models or regularization parameters. And it is true that our analysis is not based on multi-dimensional L-surfaces, but that was not the goal and is definitely beyond the scope of our work. In addition, Reviewer 2 argues that these details for ISSM are very helpful and also useful for the reproducibility of our work. Nevertheless, we try to shorten the manuscript in this direction and to highlight the main results here.
Regarding the comment on the smoothness of L-curves, we can argue that whenever our resulting L-curve had many outliers there was something wrong with the numerics or with the underlying model choices we did for those runs. So the smoothness of our L-curves gives us

a kind of measure of whether everything in the background is working properly with the model or the numerics, just as the subdomain L-curve analysis does it in more detail. In our opinion, the comparative shapes of the curves for linear and non-linear sliding are conspicuous, as they show an explicit different behavior and the steeper L-curves for non-linear sliding helps us to bracket the range from λmin to λmax smaller. Both aspects are also something observed by Wolovick et al. (2023).

I was also not particularly convinced by the comparison with proposed subglacial lake positions. Partly because it is not actually known whether lakes really do exist in these locations, but mostly that I do not see much meaningful correlation between the basal drag fields and the lake outlines in the figures.

We agree with the reviewers opinion on the lake candidates. Therefore, we will remove this section in the revised version, which was also recommended by Reviewer 2.

This manuscript appears to rely a lot on reference to Wolovick et al. (2023), but I think the most important parts of methodology should be restated here. In particular, the procedure for picking λbest, λmin and λmax, which are crucial to the results of this manuscript, is buried in an appendix in Wolovick et al. and should be stated clearly to help readers understand the L-curve figures.

We will add a description of the picking procedure of λbest, λmin and λmax to the new manuscript version.

We added:
"A basal drag inversion is performed for each sample, and the resulting costs Jobs and Jreg are recorded. To avoid the arbitrary selection of the best λbest value by hand-picking, we generate a smoothed L-curve using the 25 sample results in the (ln(Jreg), ln(Jobs)) space and determine the λbest value by calculating the maximum curvature (second derivative), following the method of Wolovick et al. (2023). The smoothing is required in this case because the second derivative tends to amplify noise. The smoothed trade-off curve is obtained by subsampling the 25 model results into 1000 logarithmically spaced λ subsamples. Subsequently, 50 different smoothing wavelengths are tested to identify the optimal one in the ln(λ) space through a sort of meta-regularization. Therefore, we calculate the variance for each wavelength based on the curvature of the smoothed curve and the scatter of the original model points. For further details on the variance calculation, see Wolovick et al. (2023). The wavelength that minimizes the sum of both normalized variances is selected. The total logarithmic curvature $d^2(\ln(J))/d^2(\ln(λ))^2$ is calculated and the location of maximum curvature, representing the sharpest corner, identifies the λbest value of the L-Curve. The curvature is also recorded when it drops to half (arbitrary value) of the maximum value, providing a complete bracketed range from the minimum acceptable λmin to the best value λbest and the maximum acceptable λmax."

Overall, I believe that this manuscript contains some very interesting and important work, but that it is somewhat hidden amongst a wide variety of ideas which could be organised in a better way, and some of which seem specific to the model being used. I think this manuscript could benefit from being edited down to a more concise form, and that some restructuring could benefit its readability.

Thank you for all the comments and suggestions. We will revise the manuscript in terms of structure and language, as well as shorten it to emphasize the important points, especially in the results section, as also recommended by Reviewer 2.

I would recommend major revisions before this manuscript is accepted for publication.

**Specific comments:**

**Abstract**

Lines 10-12: It's a bit unclear what "best" and "worst" mean in terms of L-curve behaviour. Perhaps saying Pine Island produces the smoothest curves would be a better way of wording this?

We agree with your comment and have edited the wording in the revised version.

"Pine Island Glacier exhibits the smoothest curves, and the slow-flowing areas such as Roosevelt Island reveal rather poorly shaped L-curve behavior for the basal drag inversion."

Lines 16-17: I don't agree with using "more accurate" here. The choice of parameter, as the authors point out, is a balance between matching and regularising the observations. Accuracy could be interpreted as getting the closest fit to observations, which isn't the point being made. The regularisation parameter itself cannot be defined as being accurate or not, as there is no physical real-world comparison to make.

Thank you for the remark, we have clarified the wording in the revised version.

"The analysis suggests that non-linear friction laws are preferable to linear sliding because of reduced variance of the overall inferred friction coefficient, faster convergence, as well as steeper L-curves leading to a more well-defined corner region."

"The analysis suggests that non-linear friction laws are preferable to linear sliding because of reduced variance of the overall inferred friction coefficient, as well as steeper L-curves leading to a more well-defined corner region."

Line 18: "improved performance" in terms of what? Convergence, smoothness of L-curves? This should be specified.

We get an improved performance in terms of the total model variance ratio, faster convergence and smoother L-curves when NCUAS is included. We have specified this in the revised version.

"We show that a Budd-type friction law incorporating effective pressure from a subglacial hydrology model rather than a simple geometry-based approximation achieves improved performance in our inverse model in terms of total model variance ratio, as well as faster convergence and smoother L-curves. "

Line 20: Should this be a comma rather than two separate sentences?

Yes, we have combined both sentences in the revised manuscript.

**Section 1. Introduction**

Lines 35-36: Could you expand on what exactly is meant by "the majority of high velocities are cause by sliding", or provide a reference? Is this just saying that sliding causes higher velocities than flow driven by surface gradients, or is it claiming that changes to basal sliding have a larger influence than other processes such as mass balance?

We have intended to express that the fast velocities in the ice streams are mostly caused by basal sliding and have added a reference to the work of Engelhadt and Kamp (1998) as an example. The text now reads as:

"In the context of a better understanding of ice sheet processes, it is particularly important to examine the distribution of friction at the ice-bed interface, as this process has a major influence on the ice velocity, particularly in the fast-flowing areas (e.g., Engelhardt and Kamb, 1998). Since the distribution of friction underneath the ice sheets is difficult to observe directly, ice flow models are used to determine the basal drag."

"In the context of a deeper understanding of ice sheet processes and improved initial conditions, it is particularly important to examine the distribution of friction at the ice-bed interface. This process has a major influence on the ice velocity, especially in the fast-flowing areas (e.g. Engelhardt and Kamb, 1998), but cannot be measured on large scales."

Line 43: I'd say "represent" rather than "compute". The parameters from inversions broadly represent a combination of several factors (as is pointed out in the next paragraph), rather than being a value which physically describes one thing.

We have changed that in the revised version.

Lines 87-89: This reads as if the Budd law doesn't account effective pressure. Should be reworded for clarity.

We agree, thanks for the remark. We have reworded the sentence.

"In the literature, instead of a Budd-type friction law (Budd et al., 1979), a Weertman friction law (Weertman, 1957) is often used (e.g. Morlighem et al., 2010, 2013; Joughin et al., 2004; Ranganathan et al., 2021) in which no effective pressure is taken into account."

Lines 98-100: Isn't the inverse problem in glaciological models always ill-posed, regardless of sliding laws or domains? Could this sentence be explain further?

Yes, that is true, but adding regularization to the minimization problem makes the problem of course less ill-posed. And the idea behind it is that we might have regions that are problematic to treat in the modeling, e.g., rock outcrops. If these regions have more outliers for small λ values in the L-curve, the problem may not be regularized well enough and still be ill-posed. But if it shows a good L-curve, we can assume that it is rather well-posed. Overall,

we wanted to express that some areas are less ill-posed due to regularization and others may be more ill-posed because more regularization has to be used. However, we recognize that the spelling is misleading and remove this point from the text. Also in lines 292 and 356.

**Section 2. Method**

Some restructuring in this section could be beneficial, as there appears to be some repetition, and jumping between topics.

We agree, thank you for the remark. Restructuring of this part has been done and incorporates now also changes suggested by the other Reviewer.

Lines 108-115: I don't think this section is necessary. ISSM can be introduced at the start of 2.1 instead, and other parts stated in the relevant sections.

That is true, we have deleted this paragraph and added the ISSM content into the section 2.1.

Lines 134-139: This could be moved into 2.3, with the rest of the inversion description.

We have moved this sentences into section 2.3.

Lines 166-176: The discussion of B fields seems to interrupt the description of the forward model. It could be better placed in the subglacial hydrology section as it is being discussed in relation to CUAS.

We have added this paragraph to the subglacial hydrology section.

Lines 189-194: Why not introduce both effective pressure parameterisations in the same section (ie. move this to the next section).

We have also moved this paragraph into the subglacial hydrology section.

Lines 233-236: These two explanatory sentences belong in the introduction (in fact, I think they restate something already in the introduction)

We have removed those sentences at this point and rearranged the regularization part in the introduction.

Line 243: Given how important the values of λbest, λmin and λmax are for this work, I think the method should be shown here rather than just a reference.

As mentioned before, we will add a description of the picking method for the L-curve analysis in the revised version.

Lines 276-280: Should the detail about the forward model solver be under the Forward Model section?

Yes, that is a good hint. We have added this sentences to the forward model section.

**Section 3. Results**

Lines 304-305: Why not just use a full range of [10-3 104] for all L-curves?

Yes, this would be a possible option, but we deliberately shifted the range of the L-curve further upwards using a linear sliding law in order to avoid small λ values below 10^-2 as we only obtained outlier models for such small λ values, which are not useful. This can be explained by the fact that these outlier models for small λ values reflect that a too small weighting of the regularization term increases the non-convexity of the inverse problem again. Since we are dealing with an ill-posed inverse problem adding regularization the problem gets more convex and with that easier to solve for the optimization algorithm (convex functions have only global minima, e.g. Rockafellar (1970)). But when using too little weight for the regularization term the problem is still non-convex (global and local minima), which make it again more difficult to solve.
On the other hand, we do not need more points above 10^3 for the L-curve using non-linear sliding laws, since the vertical λ-limb of all non-linear experiments are already very pronounced. However, we are particularly interested in the corner of the L-curve, i.e., extending the λ range upwards would only lead to more computing time and costs, which we would like to avoid. For the sake of simplicity, we have shifted the range for the linear sliding law upwards and kept the number of λ values the same, but still get pronounced limbs in the flat and vertical curve. Because of this, we argue that we do not need to run any further models as they would not give us any new information.

This was stated in the methods section in the manuscript and was probably disconnected from the results section. We have moved this part into the results section (see also RC2 suggestion).

Line 313: How are outliers identified? I assume these are points which lie over a certain distance from the smoothed tradeoff curve, in which case this should be specified. Or are they simply the inversions which struggled to reach convergence? In particular, for Fig. 8(d,g,h) it isn't clear why some of the outliers shown shouldn't be part of the curve.

We define inversion runs as outlier models if they are not fully converged. We further declare models as outlier if they are above a certain distance to the nearest data cost or regularization cost model by choosing a threshold value. Reviewer 2 also noticed this, so we have added a description of the definition of outliers into the revised version. Indeed, for Fig. 8(d,g,h) it is a bit unclear, we will check the algorithm again in these cases and make sure that we have used the same threshold for all experiments.

"We classify inversion runs as outlier models (gray dots in Fig. 6) if they fail to fully converge or if the cost function result exhibit non-monotonic behavior. In addition, we further declare models as outliers if they lie over a certain distance from the next data cost or regularization cost model by choosing a threshold value. "

Lines 318-329: Was the same smoothing of kinit and the same values of εgttol and Δxmin applied for all inversions for all sliding laws? Given that these can have an important impact of the results I assume a like-for-like comparison has been conducted, but it could read as if different smoothing was used in different cases.

No, this is not the case, we had to smooth kinit further for the use of non-linear sliding in our inversions. For linear sliding we therefore use only one averaging iteration, but for the non-linear sliding law, we use of three iterations. We also chose different values for εgttol and Δxmin for linear and non-linear sliding (εgttol=$10^{-3}$ and Δxmin=$10^{-5}$ for the three linear sliding experiments and εgttol=$10^{-6}$ and Δxmin= $10^{-4}$ for the three non-linear sliding experiments) to ensure convergence of the model.

We state this in the L-curve analysis part of the results section. Although a "like-for-like" comparison would be beneficial, this is not possible here, because a non-linear problem is inherently more difficult to solve than a linear one. Thus, more averaging iterations are used.

We have rephrased this as:
"We used nearest-neighbor averaging to smooth the initial drag coefficient kinit resulting from the driving stress (see Eq.(6)) for the runs with linear friction law. Unfortunately, further smoothing (three times nearest-neighbor averaging) was needed for the runs with non-linear friction to ensure convergence of all those runs. ... Admittedly, the mentioned adjustments, limit the comparability between simulations with linear and non-linear friction laws."

"Another approach to address convergence issue involves using nearest-neighbor averaging to smooth the initial drag coefficient kinit resulting from the driving stress (see Eq. (6)) for the runs with linear friction law. Unfortunately, further smoothing (three times nearest-neighbor averaging) was needed for the runs with non-linear friction to ensure convergence of all those runs. … Admittedly, the mentioned adjustments, limit the comparability between simulations with linear and non-linear friction laws."

Lines 335-342: As a more general point, is "best" the right word to use? The choice, as is always the case with L-curves, is quite subjective. I agree that the identified corner regions are a good guide for picking your regularisation parameter, and appreciate the more rigorous methodology behind picking a value rather than doing so by eye, but I would assume any pick within this region would be reasonable. Specifically, as noted here, in some cases the identified λbest is very close to λmin, and perhaps another argument could be made to say that a value halfway along the curve between λmax and λmax could be the best choice. Do you have any insight into what variability the choice within the corner region causes in the output? It would be interesting to see the differences between your λbest and a value one might pick by eye, to give more of an idea of how much this matters.

Perhaps it is not the best word, as it is of course not the best drag, because it is not the real one.

Of course any value between λmin and λmax is considered as reasonable and we state this in the text. For this reason, we have bracketed the range of acceptable λ values with λmin and λmax instead of specifying only λbest. You could also call it λoptimum (2nd derivative of the curve), but of course that wouldn't be entirely obvious either. Based on the L-curve curvature (maximum curvature of the L-curve) an alternative name could be λmax_curvature, but we decided to stick with λbest according to the work by Wolovick et al. (2023).
Without any doubt, the selection of the picking method for the λbest value is still a "modelers choice" and we make it very clear that this is just our choice.

The range of acceptable λ values includes the choice of an arbitrary "heuristic" threshold for the curvature of the L-curve. We used a value of ½ of the maximum curvature to define λmin and λmax. In terms of our selection method for the λbest value in the range of λmin and λmax, this is the "best" value. But, as we have also mentioned in the manuscript, our picking method does not match in the case of Fig. 6 a and e (m=1, Weertman and m=3, Nop experiments) with the value one would choose from a visual point of view.
As we are aware of some limitations of our picking method, we would probably consider a new method, based on an integrated curvature, in future studies. However, since we particularly address Fig.6 f (or rather the corresponding experiment and analyze λbest, where the value of our method corresponds to the one that would be selected by eye), we would not like to lengthen the manuscript any further.
All in all, Fig. 15 shows what difference it makes to analyze λmin versus λbest versus λmax.

Lines 343-345: Is it necessarily to be expected that λ values would be close to each other? What were the differences in the other inversions mentioned here which display larger variability, and why should that make them less trustworthy?

Also Reviewer 2 has remarked this point and we have to admit that we have expressed ourselves incorrectly here. We cannot associate good results with the small variation of the λ best value ranges of all experiments. In addition, the subdomain L-curve analysis shows that the λ values do not necessarily have to be in the same range for different areas, as they require different amounts of regularization. We removed this point from the revised version.

Lines 346-354 & Fig.7: The convergence could be specific to the particular inversion process of ISSM with the chosen optimisation algorithm of M1QN3. Appendix B1 of Barnes et al. (2021) shows improved performance (greater minimisation of the cost function) achieved using an interior point algorithm (Byrd et al., 1999). Do you have any thoughts on whether you might find similar differences in convergence between your cases using a different algorithm such as this?

Yes, this could depend on ISSM and the M1QN3 algorithm and other algorithms might achieve better convergence here. However, we are relying on the implemented algorithms in ISSM. And as Morlighem et al. (2013) showed, a BFGS algorithm (quasi-Newton method), as used here, performs significantly better than for example a steepest-descent algorithm. And we also recognize that the M1QN3 algorithm achieves good convergence, but there are certainly improved algorithms which one could use. But of course we cannot say whether other algorithms, like the one shown in Barnes et al., (2021), perform better without further comparisons, which is beyond the scope of our work. But for sure, we can add a sentence to that in the revised version.
But we consider that it is beneficial to keep the convergence part in the manuscript. Reviewer 2 also found the details very useful and for the community using ISSM it can be helpful and increase reproducibility.

"Overall, we have to admit that the convergence results could be specific to the particular inversion process of ISSM with the chosen optimization algorithm of M1QN3. Even if we recognize that the M1QN3 algorithm achieves good convergence, there are certainly improved algorithms which might achieve better convergence. But we cannot say whether other algorithms, like, e.g., an interior point algorithm Byrd, 1999, as shown in Barnes, 2021, perform better without further comparisons. "

Line 355: I think this section should be 3.2. It contains enough to stand alone. I like the idea of running inversions on subdomains to find whether different regularisation could be required under different physical conditions. I think the values of λbest should be highlighted on Fig.8 as they are in Fig.6, to show the difference more clearly. I would personally also be very interested to see what difference would be seen in the inversion outputs going into a forward run, comparing a spatially-varying λbest to the global value.

We have renamed this section into section 3.2 even if we have moved some of the results from this section to the non-linear versus linear sliding section after being recommended by Reviewer 2. In addition, we will add the λbest values to the Fig.8.
It is not clear what is meant by simulations based on a "spatially-varying λbest". The regional L-curves are based on the same inversion result (experiment NCUAS, m=3). It is just another post-processing step to analyze different geomorphological settings. One could do forward runs using the different λbest values and compare those runs with the "global" λbest but this is out of scope for this manuscript. As shown already in Fig. 15, higher λ values would result in more smoothing (more regularization).

If the Reviewer thinks of a setup with spatially varying amounts of regularization (λbest) based on, e.g., information from the regional L-curves for different geographical settings, this would be a very interesting idea. Unfortunately, we don't see a practical way on creating such a map, yet. (Classification by ice speed as well as surface, bed and thickness slopes?)

Fig 8: This figure, and some that follow, could be made clearer by highlighting the corner region in a different colour, rather than the thicker line currently used. I assume the circles with white in the middle are outliers, but this is not labelled on the figure. Some of the L-curves presented in this figure could benefit from an extended range of values, as the corners appear very close to the end of the intervals used.

Thank you for pointing this out. We will edit the corner regions of the L-curves accordingly in the revised version.
Correct, the white circles show outlier models and we added a sentence about these in the figure caption.
We understand the reviewers point (extending the ranges for the L-curves), but here we argue again that an extended range is not associated with new information, rather only with increased computational costs. In particular, this is only the case for Fig.8 d,e,i. Those subdomains do not benefit much from the regularization.

Figs 9,10,12: The diamonds showing location of λbest is not clear. The values of λbest could also be shown.

In the revised version, we will adjust the diamonds and add λbest values in these figures.

Lines 464-471: I'm not sure I follow the reasoning for using the same λbest in both cases. The whole point in the inversion process is surely to pick an optimal regularisation for each case, in order that the output is most suited to the individual problem. I would think it was more instructive to compare the outputs that come from each experiment's λbest value. Also, the numbers given do not correspond to the λbest values stated on Fig.6. If these are coming from different L-curves it should be made clearer.

Yes, the point of the L-curve analysis in the inversion process is to pick an optimal regularization for each case.
However, the point we intended to raise here is how Nop versus NCUAS in a Budd sliding law account for the structure of k^2, since we are discussing k^2_W=k^2_BN. If we would use different λbest values in this case, one result might be smoother than the other and no longer comparable. The goal is to show that NCUAS captures significantly more of the structure of k^2B than Nop, which for a Weertman sliding law is simply taken into account by k^2W. In the ideal example or ideal effective pressure N, k^2B would be a constant.

The selected λ values used in the figure are the nearest (discrete) samples on the L-curve. We don't have a simulation performed with the estimated λbest values at this point. See section "3.4 Best drag estimate" for the simulation using the exact λbest value. We have clarified this in the revised version.

"To overcome the issue of non-comparability with respect to varying degrees of smoothness when using different λbest values, each basal drag coefficient is evaluated at the nearest (discrete) sample of λbest of the respective NCUAS experiment."

Fig.11: Why not also show the Weertman output?

Yes, we could of course also show the Weertman output, but the aim here was to show what effect the use of an effective pressure N (NCUAS or Nop) has on the basal drag coefficient result k^2. In the case of the Weertman law, a basal drag coefficient k^2 would account for every structure that we otherwise put into the effective pressure field for the Budd friction law. Therefore, we argue that a map of k^2 for Weertman sliding provides no further insights.

Lines 519-520: Could you explain how the chosen case is the "best estimate" of those on Fig.6?

We have added a description in the revised version of why this chosen λbest is the best estimate here, as also Reviewer 2 has noted this.

We now state:
"For this particular λbest, the cost curvature (Fig.9 d) corresponding to the L-curve in Fig.6 f (NCUAS, m=3) has a clear peak at which our picking method chooses the λbest value. This would also be the position one would pick the λbest value based on visual inspection of the L-curve. "

Fig.15: This is a great figure for showing why regularisation choices are important, and the difference they can make!

Thank you!

**Section 4. Discussion**

Line 558: It may be desirable from a practical perspective, but for better results maybe picking a few different regularisation values based on ice speed or other obvious differences in the physical setting would be the way to go. I find this idea very interesting.

Yes, this is a very interesting point, as mentioned in a comment above, and definitely something to think about in the future, but this is outside the scope of this manuscript at the moment.

Added a bit about that topic into the discussion.

Figs 17,18: I don't personally find these to be particularly convincing. While there are a couple of places where the basal drag field lines up with the white lines, there are also places where the outlines cross areas of higher friction. What are the locations of these "possible" lakes based on? Is there a correlation with their positions in the effective pressure fields Nop and NCUAS? Is similar correlation seen for the Weertman inversion output as the Budd ones?

There is no particular correlation seen in the effective pressure field Nop or NCUAS with the lakes and it also does not make much sense to compare the lakes with the effective pressures, as these are not explicitly included in the determination of these fields. Also for the basal drag inversion output using the Weertman sliding law we can find similar correlations with the possible lake candidates. But, we realize that the analysis does not contribute much more to the output of the manuscript and Reviewer 2 also noted this point. Therefore, we have decided to shorten the manuscript at this point and to remove the comparison of the basal drag and the possible lake candidates from the revised version.

**References**

Barnes, J.M., Dias dos Santos, T., Goldberg, D., Gudmundsson, G.H., Morlighem, M. and De Rydt, J., 2021. The transferability of adjoint inversion products between different ice flow models. *The Cryosphere*, *15*(4), pp.1975-2000.

Byrd, R.H., Hribar, M.E. and Nocedal, J., 1999. An interior point algorithm for large-scale nonlinear programming. *SIAM Journal on Optimization*, *9*(4), pp.877-900.

Wolovick, M., Humbert, A., Kleiner, T. and Rückamp, M., 2023. Regularization and L-curves in ice sheet inverse models: a case study in the Filchner–Ronne catchment. *The Cryosphere*, *17*(12), pp.5027-5060.

**Our References**

Barnes, J.M., Dias dos Santos, T., Goldberg, D., Gudmundsson, G.H., Morlighem, M. and De Rydt, J., The transferability of adjoint inversion products between different ice flow models, *The Cryosphere*, *15*(4), pp.1975-2000, 2021.

Engelhardt, H. and Kamb, B.: Basal sliding of Ice Stream B, West Antarctica, Journal of Glaciology, 44, 223–230, 875, https://doi.org/10.3189/S0022143000002562, 1998.

Morlighem, M., H. Seroussi, E. Larour, and E. Rignot, Inversion of basal friction in Antarctica using exact and incomplete adjoints of a higher-order model, *J. Geophys. Res. Earth Surf.*, 118, 1746-1753doi:10.1002/jgrf.20125, 2013.

Rockafellar, R.T. Convex Analysis. Princeton University Press. http://www.jstor.org/stable/j.ctt14bs1ff, 1970.

Wolovick, M., Humbert, A., Kleiner, T., and Rückamp, M.: Regularization and L-curves in ice sheet inverse models: a case study in the Filchner–Ronne catchment, The Cryosphere, 17, 5027–5060, https://doi.org/10.5194/tc-17-5027-2023, 2023.

**Response to Reviewer 2**

**REVIEW: Improved basal drag of the West Antarctic Ice Sheet from L-curve analysis of inverse models utilizing subglacial hydrology simulations**

The original reviewer comments are in black and our responses are highlighted with blue color.

**Overall answer to RC2:**

Dear Reviewer,

thank you for all of your comments which will contribute to the improvement of our manuscript! We have answered all your comments in our final author comment. To facilitate the response to your comments, we have already prepared a revised version in which we already incorporated most of the specific comments.
Both reviewers suggest major revisions before publishing the manuscript. Therefore we will reorganize our manuscript and try to make the structure of the manuscript far more efficient to read, especially we for the method and results sections. As both reviews suggest to remove or condense the section on subglacial lakes we decided to remove it from the upcoming manuscript version. In addition, we will work on the presentation of our figures and in particular on the figures showing the L-curves and their cost function curvature. We will add a new part to the discussion section in which we want to discuss our results with regard to the regularized Coulomb friction law. Further, we will add a description of our selection method for the λbest, λmin and λmax of our L-curves.

The authors perform an L-curve analysis on a suite of basal drag inversions applied to a model domain that encompasses much of WAIS. L-curve analysis allows them to examine the balance between cost function terms when different degrees of regularization are utilized in the inversion, represented in the manuscript as a range of $\lambda$ values, where $\lambda$ is the weight associated with the regularization term in the inversion cost function. Inversions are analyzed with respect to the trade off between components of the cost function: Jobs, associated with velocity misfit compared to observations, and Jreg, associated with regularization. The best performing inversion is chosen based on the value of $\lambda$ which minimizes the cost function without either component of the cost function growing too large. This is done analytically by finding the maximum curvature of the L-curve.
Suites of inversions are performed under the assumption of linear (m=1) and nonlinear sliding (m=3), and using three treatments of effective pressure: the exclusion of effective pressure (Weertman), effective pressure approximated based on ice geometry (Budd with Nop), and

effective pressure calculated from a subglacial hydrology model (Budd with NCUAS). Comparisons are drawn across the resulting 6 experimental groups allowing for analysis of the $\lambda$ values which produce the best inversions when different physics are applied in the model parameterization. Additionally, the authors break their large model domain into 6 smaller subdomains that incorporate distinct geographic characteristics and perform the same analysis. This provides practical insights into how the inversion algorithm functions when applied to various portions of the ice sheet, and illustrates some complications of inverting for basal drag across large portions of the ice sheet without specialized treatment to certain areas.

**Overall Statement:**

On balance, there is a lot of very practical and useful information in this paper. The authors provide a great deal of detail with respect to the methods and L-curve analysis, which makes this work a resource for anyone interested in incorporating a more rigorous analysis to the inversion steps in their modeling procedure. This type of practical resource is not so often published in the glacier modeling community, although it perhaps should be to improve the reproducibility of modeling outputs and provide greater access to learning new methods/improving existing ones. Additionally, the main results of this work provide easy to implement improvements to basal drag inversions across a large portion of the Antarctic Ice Sheet.

Thank you for your remarks! Of course, we hope that we can improve reproducibility with the results of our manuscript and we also wish that our results will be helpful for further model approaches in the community.

My main critiques of this work pertain to the organization of the paper. I believe that the readability of the manuscript could be largely improved by reducing the length and simplifying the language, particularly in the Results section where there seems to be quite a bit of repetition and bouncing back and forth between ideas. That being said, I think the analysis of this work is sound and interesting and merits publication.

We agree that our manuscript should be further restructured as some parts are repetitive; this was also noted by Reviewer 1. Therefore, we will try to simplify the language and reduce the length, especially in the results section. A minor restructuring has already been done based on the specific comments of both reviewers.

**General comments:**

The Results section of this paper is 16 pages. While I appreciate the attention to detail, it would benefit from some simplification and reorganization. It is currently divided into subsections, but discussions pertaining to other parts of the study are present in each of the subsections. This causes quite a bit of repetition throughout the Results. I would suggest tightening up each of the subsections to highlight the main result and simplify the message. Comparison of the main results is done quite nicely later in the paper. It doesn't need to be addressed here as well. Readability of the Results section could be further increased by either condensing the description of L-curve analysis and focusing on giving clear interpretations of the figures, or by segmenting off some of these details into a supplemental

document.

Thank you for the hints. We will reorganize the manuscript in the revised version more efficiently and shorten it to put the main highlights in the focus.

The text on many of the figures is quite small, making the subplots difficult to interpret. I would suggest making some of the smaller figures larger in general; specifically figures 7, 9, 10, 12, and 16. Some of the markers on subplots are also quite difficult to see. Figures 8 and 15 both illustrate the information being presented very nicely.

We will edit all of the mentioned figures in the revised version.

The Discussion and Conclusions sections are well written and clearly move through the main conclusions of the paper. However, the connection to subglacial lakes seems a bit out of sync to the rest of the paper, and I am not entirely convinced of the argument being made. It seems to me that correlation between $\tau b$ and observed lake location is due to the use of N in the friction inversion. I think this section could probably be removed or condensed, but right now is somewhat distracting from the main results of this work.

We have analyzed the correlation of the possible lake candidates and the two effective pressure fields Nop and NCUAS used in the manuscript and could not find any significant relation. Further, we would argue that it does not make much sense to compare the effective pressure fields with the subglacial lake candidates as they are not explicitly included in the determination of these fields. In addition, we rechecked the inversion result using Weertman with the possible lake candidates and could find a similar correlation as for the Budd-type sliding law using Nop and NCUAS. Nevertheless, Reviewer 1 also did not find the results particularly convincing either and we agree that the output does not contribute much to the overall outcome of the manuscript as well as we are aware that it is not clear whether these sublgacial lakes really exist. Therefore, we have decided to remove this section on the comparison to subglacial lakes in the revised version and thus shorten the manuscript at this point.

I would be curious to know how these results might differ when using different forms of friction laws. Weertman and Budd are both power laws, but a regularized Coulomb friction law may produce significantly different behaviors, especially when comparing the performance of the inversion across subdomains. I understand that doing the full analysis using additional friction laws is a large amount of work, and I think the results presented here warrant publication without additional modeling. However, if the authors have a sense of how these results might translate to other friction laws, it would be an interesting point of discussion.

Indeed, the regularized Coulomb friction law was not mentioned or discussed in the previous version of the manuscript. As we now apply a model for the subglacial hydrology, we get some information about the basal water pressure distribution. The regularized Coulomb law (non-linear with respect to the effective pressure) would allow for an even higher dependency of the basal motion on the hydraulic conditions. Duo to constrains on resources we can not further investigate this with additional model runs, unfortunately. We add a few sentences to the discussion and reference the work of Schoof (2005) and Joughin et al., (2019) on that subject.

"Joughin et al. (2019) recommend using a Coulomb friction law, which accounts for cavitation effects on sliding (Schoof, 2005), as it improved simulation fidelity, especially for Pine Island Glacier. Notably, Joughin et al. (2019, Fig. 1) illustrates that non-linear power laws, like the m = 3 used in our study, approximate the regularized Coulomb friction law for m > 1. Consistent with Wolovick et al. (2023) and our results, that non-linear sliding laws improve model performance. However, incorporating subglacial hydrology provides better insights into basal water pressure, and a regularized Coulomb law, which is non-linear with respect to effective pressure, could further enhance the dependency of basal motion on hydraulic conditions. Therefore, adopting the Coulomb friction law from Joughin et al. (2019), which accommodates both weak till and hard bedrock, alongside effective pressure from the CUAS-MPI hydrology model, could improve accuracy in future studies."

**Specific comments:**

**Abstract**

Line 2-5: This sentence is overly complex. I would suggest pairing it down as it is just context.

Thank you for the suggestion, we have implemented this.

"To accurately predict future glacier evolution, precise ice sheet models are essential."

Line 10: It reveals that Pine Island Glacier being → Pine Island Glacier is

We have changed the sentence accordingly.

Has been removed.

Line 11: "best" and "worst" feel like odd descriptors here without more context for the interpretation of the L-curves. Consider changing or adding a bit more detail.

Thank you for the hint, also the Reviewer 1 has commented on this. We have edited the wording in the revised version.

"Pine Island Glacier exhibits the smoothest curves, and the slow-flowing areas such as Roosevelt Island reveal rather poorly shaped L-curve behavior for the basal drag inversion."

Line 12: Remove the comma.

Done.

Line 19: "a larger part of the spatial basal drag coefficient structure" is somewhat confusing, consider rephrasing.

We have rephrased this sentence in the revised version.

"Further comparison reveals that the basal drag coefficient field has a less variable spatial structure when an effective pressure from the hydrology model is used instead of a

parameterized effective pressure, allowing us to interpret the inverted drag coefficient more precisely in terms of the basal properties rather than the basal hydrology."

Line 20: "Allowing the inverted drag…" This is an incomplete sentence, combine with previous or remove.

That is true, we have combined both sentences.

Line 21: "reflect actual variations in basal properties" implies comparison to observations. If this is what you intended it should be stated more explicitly, but I don't believe that was part of the study.

With this sentence we intended to say that when using a Budd-type sliding law with effective pressure of a hydrology model, the part of the basal hydrology gets covered by this effective pressure and the basal drag coefficient is smoothed out as it needs to take less structure into account (see Figure 11). For this reason, we can again interpret the basal drag coefficient as a kind of physical description of the basal properties instead of the basal hydrology and thus the basal drag coefficient only reflects the actual variability of the basal properties. However, we have changed the sentence to clarify things.

"…, allowing us to interpret the inverted drag coefficient more precisely in terms of the basal properties rather than the basal hydrology."

**Section 1. Introduction**

Line 30-31: "The ongoing continuous…" This sentence is awkward and reads as an incomplete thought, consider rephrasing.

We have reformulated this sentence in the revised version.

"The continuous improvement of ice sheet models (e.g. Blatter et al. (2010), Seroussi et al. (2019)) through our increased knowledge from research reduces uncertainties in future predictions."

"The ongoing development of ice sheet models (e.g., Blatter et al. (2010), Seroussi et al. (2019)) is driven by advancements in research and the resulting improved understanding of ice sheet behavior. These improvements enhance our insights into key processes, such as ice dynamics and the ongoing melting. Consequently, more accurate initial states for ice sheet models can be established, further minimizing uncertainties in future projections (Seroussi et al. (2019))."

Lines 35-45: This paragraph jumps from discussing friction to subglacial hydrology to basal drag inversions. It feels a bit jumbled. I suggest reorganizing to discuss basal friction, the difficulty of parameterizing it, and inversions together first. Then introduce subglacial hydrology as an additional complication in determining basal friction.

Thanks for the remark. We have reorganized this part of the manuscript accordingly.

Line 58: "However, it would be possible…" Suggest removing this sentence. It is not

necessary to make your point.

That is true, we have deleted this sentence in the revised version.

Line 61-70: This paragraph is good, but it should come earlier. Here you are introducing concepts that are referenced with less context in the previous paragraph as though it is the first time they are being brought up.

We have followed your suggestion and included this paragraph at an earlier stage in the manuscript.

The first two paragraphs of the introduction are somewhat disorganized and confusing in the way they introduce the main concepts of this work. But the rest of the introduction flows quite well, is easy to follow, and sets up the objectives of the paper nicely.

Thank you for the helpful comment on that and as mentioned above we have reorganized the first paragraphs of the introduction.

**Section 2. Method**

Line 118: Remove the comma.

Done.

Line 156: "model that models" is repetitive; this sentence should be rephrased.

Thanks for the hint, we have reworded this sentence.

"We use a forward model that represents the ice dynamics in an approximated way and is explained by the following higher-order Blatter-Pattyn approximation."

"We use a forward model that represents the ice dynamics in an approximated way and is explained by the following higher-order Blatter-Pattyn approximation (HO; Blatter, 1995, Pattyn, 2003) that computes the horizontal ice velocities $v\_x, v\_y$. "

Line 219: "we set it to ice pressure pi and neglect": I assume that "it" is in reference here to effective pressure, but this sentence would be clearer if that was stated explicitly.

Yes, we agree on that and have changed it in the revised version.

"Since Thurston Island is not included in the CUAS-MPI model domain, we set the effective pressure at this site to ice pressure $p\_i$ and neglect the water pressure $p\_w$ due to the predominating low velocities at this location."

Line 237: To improve this instability of the problem, it is beneficial to regularize it → To improve the instability of the inverse problem, it is beneficial to add regularization.

Thanks for the comment, we have rephrased the sentence accordingly.

In general, frequent use of pronouns in technical writing adds confusion. When you can be specific in what you are referencing, it is good to do so.

Thank you for the hint, we will take care of this in the future.

Line 244-246: Description of $\lambda$ range should be simplified. The distinction of different ranges being used is not necessary here and adds confusion because it is not yet stated when or why each range is applied.

Yes, we have shorten this paragraph in the manuscript and generalized the description of the different λ ranges and will just mention them in the according results section.

"For the L-curve analysis, we use ranges of λ in [10^-3,10^4], as well as 25 logarithmically spaced samples. For every sample, a basal drag inversion is performed. At very small λ values we observed that no convergence is achieved for linear friction laws. From a mathematical point of view, this is reasonable, as the problem is again non-convex due to the non-linearity of the forward model (Eq.(1)) and the regularization contributes too little to the cost function."

"For the L-curve analysis, we consider ranges of λ in [10^-3,10^4] with 25 logarithmically spaced samples. A basal drag inversion is performed for each sample, and the resulting costs J_obs and J_reg are recorded."

Line 246-249: Not sure that this is necessary.

As mentioned in the previous comment, we deleted the detailed description of λ ranges and just state the details in the results section.

There is a lot of really useful detail provided in this section that is often left out of modeling papers. I appreciate the inclusion of helpful tips that others in this field might find useful. But I do think that some reorganization could improve the readability and flow of ideas.

Thank you for the useful comments and hints, we have tried to reorganize the method section in the revised version with the help of your comments.

**Section 3. Results**

Line 313: How are outliers defined?

We define inversion runs as outlier models if they are not fully converged. In addition, we declare models as outlier if they lie over a certain distance from the next data cost or regularization cost model by choosing a threshold value. That is also something Reviewer 1 asked for and we have added a description of how we define outliers into the revised version.

Line 315-317: Confusing sentence, rephrase.

We have rephrased it in the revised manuscript.

"These outlier models probably arise due to the fact that the regularization term is weighted too low and the misfit term too high when using small λ values. The purpose of regularization is to make the ill-posed inverse problem with many local minima more convex, since convex functions have only a global minimum (Rockafaeller (1970)). This helps the optimization algorithm to avoid bad solutions or to find an optimal solution. However, if we regularize the problem too little (small λ values), it remains non-convex and outlier models occur."

"These outlier models probably arise due to the fact that the regularization term is weighted too low and the misfit term too high at small λ values. Regularization is essential for transforming the ill-posed inverse problem, characterized by numerous local minima, into a more convex form, as convex functions have only a single global minimum (Rockafellar (1970)). This helps the optimization algorithm to avoid bad solutions and achieve an optimal solution. However, insufficient regularization (small λ values) leaves the problem non-convex, increasing the likelihood of outlier models."

Line 319: The information about smoothing and convergence criteria in this paragraph is really useful!

Thank you for the comment!

Line 330: Maybe point to two subplot panels here that highlight the distinction being made by a very flat L-curve.

Yes, that is a good idea. We have added references to Figure 6 in the revised version.

Line 335: curve fitting procedure is → curve fitting procedure, is

Done.

Line 338: "cost curvature" is a bit confusing. I suggest rephrasing to "curvature of the cost function" at least the first time it is mentioned.

Of course, we have edited it in the revised manuscript.

Line 340: In this method, you are picking $\lambda$best to be the $\lambda$ corresponding to the absolute maximum curvature of the L-curve, if I am understanding correctly. If so, I would say so rather than describing the small side peak in the curvature figures.

Yes, that is right, we pick the λbest at the maximum curvature. But we think it is important to also mention that this method has a disadvantage, as it occasionally leads to choosing a peak value that may not really reflect the λbest value, which we might have picked by hand as it is not exactly in the corner of the L-curve (compare Fig. 6 a,e, Fig.9 c and Fig. 10 b). This is not a direct problem, as the λbest values are still in our acceptable λ range of λmin and λmax. Nevertheless, we have edited the description of it in the revised version.

"In these two figures the total curvature has a broad region of generally high curvature representing the visual corner and with a narrow small peak at the side describing the maximum curvature. As our method relies on picking the λbest value at the maximum

curvature of the cost function, this side peak is selected instead of the mean value of this broader region, which explains why the λbest is not selected exactly in the corner of the L-curve."

"Looking at Fig. 9b-d and Fig.10b-d representing the corresponding curvature of the total cost function (gray line) to Fig. 6a,e, a broad region of high curvature representing the visual corner with a narrow side peak (describing the maximum curvature) leads to selecting the latter as λbest. This results in a slight deviation from the mean of the broader region and, consequently, from the visual corner."

Line 344: Suggest removing reference to results that are not included in this work or published elsewhere.

Yes, that is a good suggestion, we have deleted this reference.

Line 345: How are you defining good results? Is this in reference to the $\lambda$best range being small? If so, can you provide some citations to other work to back up this claim?

We have to admit that we have expressed ourselves in a misleading way here. We cannot associate good results with the small variation of the λbest value range of all experiments. We define good results rather by the fact, that we obtain very smooth Lcurves for all six experiments, showing few up to no outliers. In addition, our picking method for the λbest value seems to be trustworthy as it usually picks the optimal value that one would probably also choose by hand, i.e. exactly in the corner of the L-curve. This means that our trust does not depend on the small range of λbest that we record from the experiments this could be of course more important when looking at different regions like we do in the subdomain L-curves.
As mentioned in the comment above, we will remove the sentence regarding the references not shown in the manuscript. Furthermore, we will remove the sentence about the λbest ranges and instead justify the trustworthiness of our results in a better way.

"Overall, the results of the six experiments, which consistently show smooth L-curves with few or no outliers, give us confidence that our model procedure is trustworthy. Furthermore, our method for selecting the λbest value appears to be reliable, as it generally selects the optimal value that would probably be picked by hand, i.e., exactly in the corner of the L-curve."

Deleted in the revised version.

Line 374: Fig. 8b-i → Fig. 8B

Done.

Line 379: Remove comma

Done.

Line 392: Fig. 8c → Fig. 8D

Done.

For a better orientation regarding the used subdomains, we will add the names of the different domains in the figure Fig.8 b-i.

Line 398: Conspicuous is that the subdomain → It is conspicuous that the subdomain

We have changed the sentence in the revised version.

Line 399: This sentence does not follow from the previous one.

In the previous sentence we write that the areas with many rock outcrops results in smooth L-curves (linear and non-linear). In the following sentence, we then argue that this gives us confidence in the treatment of rock outcrops in our model procedure. As we made the experience, that modeling rock outcrops often causes problems. So unfortunately, we do not understand why the sentence should not follow from this.

Line 401: I suggest separating the discussion of different geographies and discussion of different friction laws. These last few paragraphs could be moved into the section dedicated to linear vs non-linear sliding.

That is a helpful comment, thank you for the hint. We have moved those paragraphs to the 'Non-linear vs. linear sliding section' in the revised version.

Line 404: I suggest rephrasing this sentence for clarity.

We have rewrite this sentence for clarity in the revised manuscript.

"In general, this result is consistent with the results we obtained for the six L-curve experiments we made for the entire domain (compare Fig.6). Because it can also be recognized that the outliers in the linear L-curves (Fig. 6a-c) only occur for very small λ values and the L-curves with non-linear sliding law (Fig. 6d-f) only show outliers for larger λ"

Line 408: Is there really no recognizable impact? I'm not sure that I agree. There are few outliers with linear sliding, but the different regions have quite different shaped L-curves with different $\lambda$best values. Please elaborate to make this point more clear.

Yes, it is of course true that every regional L-curve has an influence on the global L-curve distribution as the cost function and the regularization cost are based on every local point in the domain and we do not calculate new L-curves for the subdomains. And with that the different local regions have of course an influence on the shapes of the L-curve and on the λbest values. We agree that we have expressed ourselves vaguely here and will rewrite the sentence in the revised version.

Table 1: It would be helpful to have the $\lambda$best values included here, or in their own table. These values are referenced quite a lot throughout the paper, but they are not listed anywhere.

That is true and a good point. We have added the $\lambda$best values to the Table 1.

Line 433: You state that $\lambda$best for the NCUAS run is located between the two other runs, but this is not clear on the figure. Please provide $\lambda$best for all three runs to make this point.

Yes, we have added the $\lambda$best values of both experiments into the text of the revised version and we will also change the outlier markers to make them more visible in the figures.

Has been deleted in the manuscript.

Figures 9 and 10: Subplots are very difficult to read. Text is too small and outlier markers are difficult to see. I would also suggest using colors that will contrast in grayscale.

This was also suggested from the first Reviewer. We will change the text size, the marker color, as well as the size and edit the used colors in those figures for the revised version.

Line 452: Unclear what "accounting for less structure" means.

What we are trying to say here is that with a positive correlation between k^2_W and N, the inversion does not need to produce much structure in k^2_B to match the observations, since most of the structure is already contained in N. This means that there will be less structure/variability in the k^2_B field and it will be smoother. The goal in an ideal world would be, that N accounts for all of the variation in k^2_W and k^2_B becomes constant for a specific region (subglacial substrate, geology) where we do not expect changes on small length scales.
For clarity, we have reformulated the sentence in the revised version.

"Conversely, a high linear and positive correlation between N and k^2_W would result in a smooth k^2_B field with less structure. Due to the reason that the inversion does not need to produce much structure in k^2_B to match the observations, as most of the variability is already contained in the N field. "

"A strong linear and positive correlation between N and k^2_W results in a smooth k^2_B field, as most variability is already captured by N, while a weak correlation necessitates more structure in k^2_B to fit observations."

Figure 11: This is super helpful!

Thank you!

Line 494: parallel to → alongside

We have changed that in the manuscript.

Line 497/Figure 12: Why not show this on the figure too?
We would decide against showing all L-curves in this figure in order to keep the design of the figure as simple as possible, because Fig. 9 and Fig. 10 already show all L-curves of our six experiments. We feel that this NCUAS example already makes it apparent that the linear L-curve is flatter than the non-linear L-curve. Nevertheless, we have reformulated the sentence to make clear that we only show this one example in Fig. 12 and we have added another sentence to express that this also applies to the other experiments.

"The results in Fig. 12 show steeper L-curves for both cases of non-linear sliding than for the L-curve result using the linear friction law. All four other experiments also exhibit this different shape behavior between linear (Fig.10 a-c) and non-linear friction law (Fig.10 d-f)."

Line 499-507: The point being made here is a bit unclear.

We will reformulate the paragraph in the revised version to clarify things.

Figure 15: Great figure!

Thank you!

Section 3.4: In this section, I'm still wondering if $\lambda$best is actually the best choice of $\lambda$, especially since $\lambda$min has less error from velocity misfit while still incorporating regularization. I think this section would benefit from reiterating the justification of the choice of $\lambda$best directly.

Well, it is not surprising that the error of velocity misfit for $\lambda$min is still smaller then the one of the $\lambda$best value, because if a smaller value is chosen for $\lambda$min the regularization term is weighted lower than the velocity misfit term. This means that minimization of the misfit term is clearly in the focus and the error is therefore smaller than if a higher weight such as $\lambda$best is used for the regularization term.
But of course, we have included a few sentences in the revised version explaining why we decided that $\lambda$best using NCUAS is the best choice in our opinion.

"For this particular λbest, the cost curvature (Fig.9 d) corresponding to the L-curve in Fig.6 f (NCUAS, m=3) has a clear peak at which our picking method chooses the λbest value. This would also be the position one would pick the λbest value based on visual inspection of the L-curve. Of course, the resulting velocity misfit using $\lambda$min produces a smaller error (Fig.15g versus Fig.15h), but this is the intended purpose of an L-curve method and λbest is still the optimal trade-off between the weights of both cost function terms. Furthermore, we can emphasize our choice by showing in section 3.2 that the inclusion of hydrology model results such as N_CUAS in the Budd friction law leads to a faster convergence (Fig.7b), as well as to improvements of the resulting fields in terms of the reduction in total variance ratio (Table 1, column 9) and in variance of k^2 (Table 1, column 2). Furthermore, we can point out in the section 3.3 that the use of non-linear friction laws is advantageous, which is also reflected by the reduction of the total variance of the derived basal drag parameter by half (Table 1, ninth column)."

**Section 4. Discussion**

Line 557: "the problem of outliers for too small $\lambda$ values is due to the fact of non-convexity" I'm unclear about what this means.

As mentioned already in a comment above, we want to point out that from a mathematical point of view the first term of the inverse problem is probably non-convex and regularization helps to make the overall problem more convex and thus solvable (convex functions have

only a global minimum, not local ones in which the optimization algorithm can get stuck). However, if the regularization term is weighted too low, as in the case of small λ values, the problem remains non-convex and is hard to solve, which ends up in outlier models. We can rewrite this sentence for clarity in the revised version.

"As already mentioned, the occurrence of outlier models at small λ values is probably related to the non-convexity of the inverse problem due to poor regularization, which makes it difficult for the optimization algorithm to find a global minimum. "

Line 562-565: Unclear about how HO is being applied- aren't your experiments using SSA inversions? Where does the HO model come in?

No, we're using the higher-order equations from Blatter-Pattyn instead of SSA (see also Eq. (1) in the method section). We also did some experiments with SSA equations, but they are not shown in this manuscript.

Line 618: could also caused → could also be caused

We have edited this in the revised version.

Line 654: As a inclusion → As an inclusion

Done.

The discussion is nicely constructed and overall very clear.

Thank you for this comment!

**Section 5. Conclusions**

Great well written summary!

Thank you!

**Our References**

Joughin, I., Smith, B. E., & Schoof, C. G. (2019). Regularized Coulomb friction laws for ice sheet sliding: Application to Pine Island Glacier, Antarctica. Geophysical Research Letters, 46, 4764–4771. https://doi.org/10.1029/2019GL082526

Rockafellar, R.T. (1970). Convex Analysis. Princeton University Press. http://www.jstor.org/stable/j.ctt14bs1ff

Schoof, C. (2005). The effect of cavitation on glacier sliding. Proceedings of the Royal Society A: Mathematical, Physical and Engineering Sciences, 461(2055), 609–627. https://doi.org/10.1098/rspa.2004.1350

---

## Author Response (AR2)

**Response to Editor and Reviewers**

Dear Felicity McCormack,

thank you for your comments. We have submitted a next revised version of our manuscript and explicitly incorporate your comments and those of Reviewer 1. Below we summarize the responses to your comments and the reviewer's comments. The original comments are in black and all our responses are highlighted with blue color.

**Response to Editor**

- L8: "... optimal trade-off between the cost function terms, resulting in smooth L-curves." → "... optimal trade-off between the cost function terms that result in smooth L-curves."
We have changed this accordingly in the revised manuscript.

- L10: "curves" → "L-curves"
Done.

- L44: I'm not sure what "... in the most realistic sense" means here? Do you mean "... in realistic settings"?
Yes, thanks for the comment. We have rephrased it as "realistic settings".

- L49: "It is therefore desirable to resemble the subglacial water pressure on a more realistic and physical basis…". Consider citing papers by Dow, Flowers, Werner, etc. here, who have made substantive contributions to the development of subglacial hydrology models towards this aim. Also Ehrenfeucht et al. (2024; https://doi.org/10.1029/2024GL111386), which describes whole-Antarctic subglacial hydrology model outputs using GlaDS.
That is a good point. We have added the citations mentioned to the new version.

- L79: numerical instability?
Yes, we have clarified this in the revised version.

- L93: "physical-based" → "physically-based"
Done.

- L230: Could you add a sentence describing why you don't use a logarithmic misfit term in the cost function equation (5), as per other studies (e.g., Morlighem et al., 2013)? This could be helpful for other modellers.
We have already discussed this topic in the discussion, but have added the following sentence in this section:

"We restrict our cost function to an absolute misfit term and opt against using a logarithmic misfit, as applied by Morlighem et al. (2013), since we do not perform a rheological inversion (see Sect. 4 for details)."

- Figure 7: update the font size of axis labels in panels b-i to be approximately the same as that of the main text
We have changed the size of the axis accordingly.

- L625: "...a reduced skin drag. Indicating, that…" → "...a reduced skin drag, indicating that…"
Done.

- A general revision of the Conclusion section for improved clarity would be good. E.g., L659-660: "The L-curve analysis taught us…" → "We find that ill-shaped L-curves with many outliers are most likely the result of inconsistencies in the model setup that should be addressed." There are also some incomplete sentences to reword (e.g., L663 starting with "Overall, …")
Thank your for pointing this out. We have edited the sentences mentioned and reworded some other parts of the conclusion.

**General things to check:**
- An overall check for grammar throughout the manuscript would be good. E.g., checking commas to delete (e.g., end of L63; L70 "examine both,"; L82 "point out,"; L86 "invert for both,"; etc.); prepositions (L119: "...on a manageable" → "...to a manageable"); and conjugations (e.g., L201-202 30% of the domain is frozen to the bed)
We have checked the grammar of the manuscript as well as commas, prepositions and conjugations throughout the text.

- Check parentheses in citations. E.g., L95 should be (CUAS-MPI; Beyer et al., 2018; Fischler et al., 2023). Similar elsewhere.
We have checked the parentheses and changed them in the revised version.

**Response to Reviewer 1**

Line 93: "physically based"
Done.

Line 178: I'd suggest removing "geometry data like" and "as well as"
Done.

Line 202: "is frozen"
Done.

Line 225: "as the target"
Done.

Line 296: "exhibits"
Done.

Line 494: If you're going to specify "modeled surface velocities, vs" this should probably be done the first time they're referred to in the sentence.
That is a good point, we have edited it in the revised version.

Line 549: Presumably should be "(Table 1, column eight)"
Done.

Line 572: "along with that of" … "demonstrates"
Done.

Line 658-9: "We conclude … smooth ones" This sentence is not very clear, and could do with rewording
Thanks for the remark. We have reformulated this sentence as follows:

"We developed a strategy to handle poorly shaped L-curves when weighting the regularization term and achieved six well-defined, smooth curves."

Line 663 "Overall, we highlight the importance…"
Done, but we have edited the complete sentence.